# Dual-color optical activation and suppression of neurons with high temporal precision

**Noëmie Mermet-Joret**[1,2,3], **Andrea Moreno**[1,2,3], **Agnieszka Zbela**[4], **Milad Nazari**[1,2,3], **Bárður Eyjólfsson Ellendersen**[1,2], **Raquel Comaposada Baro**[5], **Nathalie Krauth**[1,2,3‡], **Anne von Philipsborn**[1,2§], **Andreas Toft Sørensen**[5], **Joaquin Piriz**[6#], **John Yu-luen Lin**[4*†], **Sadegh Nabavi**[1,2,3*†]

[1]DANDRITE, The Danish Research Institute of Translational Neuroscience, Aarhus, Denmark; [2]Department of Molecular Biology and Genetics, Aarhus University, Aarhus, Denmark; [3]Center for Proteins in Memory - PROMEMO, Danish National Research Foundation, Aarhus, Denmark; [4]Tasmanian School of Medicine, College of Health and Medicine, University of Tasmania, Tasmania, Australia; [5]Department of Neuroscience, Faculty of Health and Medical Sciences, University of Copenhagen, Copenhagen, Denmark; [6]Instituto de Fisiología Biología Molecular y Neurociencias (IFIBYNE), Universidad de Buenos Aires, CONICET, Buenos Aires, Argentina

*For correspondence:
john.lin@utas.edu.au (JY-lL);
snabavi@dandrite.au.dk (SN)

†These authors contributed
equally to this work

Present address: ‡Novo
Nordisk Foundation Center
for Basic Metabolic Research,
Copenhagen University,
Copenhagen, Denmark and
Institute for Neuroscience,
Copenhagen University,
Copenhagen, Denmark;
§Université de Fribourg, Fribourg,
Switzerland; #Achucarro Basque
Center for Neuroscience, Leioa,
Spain, Ikerbasque - Basque
Foundation for Science, Bilbao,
Spain

Competing interest: The authors
declare that no competing
interests exist.

Reviewing Editor: Marcel P
Goldschen-Ohm, University of
Texas at Austin, United States

## eLife Assessment

This study develops **useful** tools for distinct optogenetic control of neuronal activity by red or blue light. The basic characterization of the activation of a red-shifted channelrhodopsin paired with a blue-light sensitive anion channel engineered to obtain desired inhibitory current kinetics is **solid**. However, evidence for their practical use under simultaneous multi-color or high frequency stimulation in cells are missing.

**Abstract** A well-known phenomenon in the optogenetic toolbox is that all light-gated ion channels, including red-shifted channelrhodopsins (ChRs), are activated by blue light, whereas blue-shifted ChRs are minimally responsive to longer wavelengths. Here, we took advantage of this feature to create a system which allows high-frequency activation of neurons with pulses of red light, while permitting the suppression of action potentials (APs) with millisecond precision by blue light. We achieved this by pairing an ultrafast red-shifted ChR with a blue light-sensitive anion channel of appropriately matching kinetics. This required screening several anion-selective ChRs, followed by a model-based mutagenesis strategy to optimize their kinetics and light spectra. Slice electrophysiology in the hippocampus as well as behavioral inspection of vibrissa movement demonstrate a minimal excitation from blue light. Of significant potential value, in contrast to existing tools, the system we introduce here allows high-frequency optogenetic excitation of neurons with red light, while blue light suppression of APs is confined within the duration of the light pulse.

## Introduction

Despite the short history of optogenetics, optical manipulation of neuronal activity has become an indispensable tool in the studies for functional analysis of brain circuits. The increasing needs for a more precise temporal and spatial manipulation of the neurons along with the untapped potentials of

optogenetics have been driving forces in expanding the repertoire of the optogenetic toolbox. Main development among these expansions is the discovery of opsins of diverse excitation spectra, with the current excitatory opsins showing light sensitivity ranging from near UV to near far-red spectra (*Yizhar et al., 2011*; *Deisseroth and Hegemann, 2017*; *Marshel et al., 2019*).

A major concern in the use of opsins with longer excitation spectra is that all opsins, in addition to their own excitation spectral peak, are activated by blue light, a problem known in the field as cross-talk or bleed-through. Attempts to create blue light-insensitive opsins will likely be unsuccessful due to the intrinsic blue absorption by the retinal molecule, the non-encoded chromophore, which captures light energy in channelrhodopsins (*Waddell et al., 1973*; *Wang et al., 2012*).

There have been several standouts attempts to overcome the problem of cross-talk. One uses a variant of blue-shifted ChR with fast kinetics in combination with a slow kinetics red-shifted ChR (R-ChR) (*Klapoetke et al., 2014*).The reasoning is that short pulses of blue light will activate only fast ChR. Since it requires a tight regulation of the protein expression as well as the light intensity, this method has not been adopted for in vivo application. Another approach is to expose axons expressing an R-ChR to prolonged photostimulation, during which axons become unresponsive to light pulses and cease to fire (*Hooks et al., 2015*; *Faress et al., 2024*). This method is limited to axonal regions as prolonged optical stimulation of the soma induces trains of action potentials (APs). Additionally, prolonged stimulation of the axons of neurons with slow spike-frequency adaptation, such as fast-spiking interneurons, may result in repetitive activity rather than depolarization block (*Hooks et al., 2015*).

A potential solution to the problem of cross-talk is to co-express a blue-shifted anion channelrhodopsin (B-ACR) along with the R-ChR to 'suppress' the excitation of R-ChR by blue light. The intention is to have a neuron co-expressing such opsins selectively produce APs to red light and not blue light. A recent study achieved this, by pairing GtACR2 and Chrimson (*Vierock et al., 2021*). Due to its relatively slow kinetics, this system is more suitable for optical manipulation of lower frequency stimulation. Additionally, optical activation of these opsins, which have a long opening lifetime, may elevate the concentration of intracellular ions with unintended physiological complications (see Discussion).

For these reasons, we created a system with fast closing kinetics. We chose vfChrimson – currently the fastest R-ChR – to gain precise temporal control of red-light excitation (*Mager et al., 2018*). To generate a B-ACR with vfChrimson matching kinetics we used a model-based mutagenesis approach to modify the kinetic properties of ZipACR, an ultrafast blue light-activated chloride channel (*Govorunova et al., 2017*). It must be noted that due to the high intracellular chloride concentration in the axonal terminal (*Turecek and Trussell, 2001*; *Messier et al., 2018*) the use of a chloride channel-based expression system must be restricted to the somato-dendritic regions (*Mahn et al., 2016*; *Messier et al., 2018*).

We describe several variants of ZipACR, focusing on two, I151T and I151V, which display appropriate kinetics, light spectra, and photocurrent magnitudes for the purposes of our co-expressing system. We then paired the optimized anion channels with a membrane-trafficking optimized vfChrimson (IvfChr) in a single expression unit. The new system, Zip-IvfChr, when expressed in neurons, can drive time-locked high-frequency APs in response to red pulses (635 nm). On the other hand, in response to blue pulses (470 nm), Zip-IvfChr produces no APs.

As expected, blue light pulses transiently suppress APs in Zip-IvfChr-expressing neurons, but the suppression is fully reversed within 5 ms after termination of light pulses due to the use of fast ZipACR mutants. When tested in the facial motor nucleus in the brainstem, Zip-IvfChr activation by red but not blue light triggered vibrissa movement of large amplitudes. Due to its high temporal precision of activation and suppression, this tool could be further developed for independent optical excitation of distinct neural populations.

## Results

### The strategy to reduce blue light-mediated excitation by red-shifted channelrhodopsins

We initially tested our strategy using existing variants of excitatory and inhibitory channelrhodopsin. The channelrhodopsin Chrimson and its variants have the most red-shifted spectra (*Klapoetke et al., 2014*), and their response to 470 nm has a slower on-rate compared to the 590 nm light

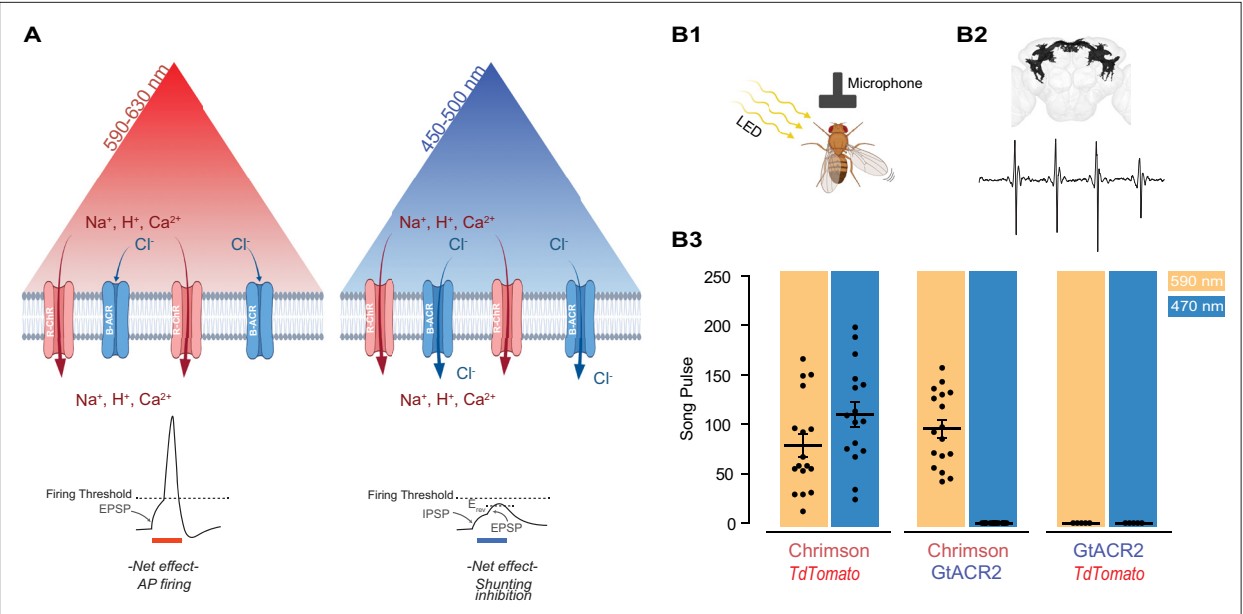

**Figure 1.** Reducing blue light-mediated excitation in red-shifted channelrhodopsins – proof of concept. (**A**) Schematic representation of the approach. A red-shifted channelrhodopsin (R-ChR) is co-expressed with a blue-shifted anion channel (B-ACR). When red light is ON, there is an overall excitation of the cell due to the more dominant R-ChR response. When blue light is ON, the shunting inhibitory effect of B-ACR reduces the excitation induced by R-ChR to blue light. Created with BioRender. (**B**) Validation of the approach in flies expressing Chrimson, Chrimson and GtACR2, or GtACR2 alone. (**B1**) Experimental setup to record the courtship song of solitary male flies. Created with BioRender. (**B2**) Top: Example of a reconstructed neuronal arborization of P1 neurons, Bottom: example of a typical courtship song induced by LED pulses. (**B3**) Courtship song production of the solitary male flies during 10 s of constant illumination with 590 nm (amber) or 470 nm (blue) light. Each dot represents an individual *Drosophila* (mean ± SEM, n=5-18).

The online version of this article includes the following figure supplement(s) for figure 1:

**Figure supplement 1.** Red-shifted ChR variants are activated by 470 and 590 nm pulses of light.

(*Figure 1—figure supplement 1A , B*). Despite this, 470 nm light is equally effective in inducing APs (*Figure 1—figure supplement 1C*). To overcome this limitation, we decided to take advantage of blue light-gated inhibitory chloride channels which effectively block AP upon exposure to blue light (*Govorunova et al., 2015*; *Govorunova et al., 2017*). By co-expressing the B-ACR variant GtACR2, along with a Chrimson variant, photostimulation of targeted neurons by red light would be expected to only activate R-ChR, which causes photoactivation of these neurons. Blue light, however, activates R-ChR as well as GtACR2. This allows the chloride channel to induce shunting inhibition of the neurons which in turn would be expected to prevent firing of the neurons by blue light (*Figure 1A*).

## Proof of concept of the co-expression strategy in *Drosophila*

As the first step to test the potential validity of this approach, we took advantage of the well-studied male-specific P1 neurons in the fly central brain that elicit courtship singing upon activation (*von Philipsborn et al., 2011*; *Inagaki et al., 2014*; *Ellendersen and von Philipsborn, 2017*; *Mohammad et al., 2017*; *Figure 1B1, B2*). When Chrimson was co-expressed with an inactive control protein (tdTomato), illumination at 470 and 590 nm wavelengths led to similar amounts of courtship song (110.1 ± 12.8 and 78.5 ± 11.6 pulses per 10 s, *n* = 16–17, respectively), demonstrating that blue light at the chosen intensity can efficiently activate Chrimson (*Figure 1B3*). When Chrimson was co-expressed with the blue-sensitive anion channelrhodopsin GtACR2, 590 nm light illumination reliably elicited male courtship song (95.2 ± 9.14 pulses per 10 s); while 470 nm light illumination did not evoke any detectable pulse song (*n* = 17–18). Neither 590 nor 470 nm light pulses exposure induced courtship song in flies expressing only GtACR2 (*n* = 5).

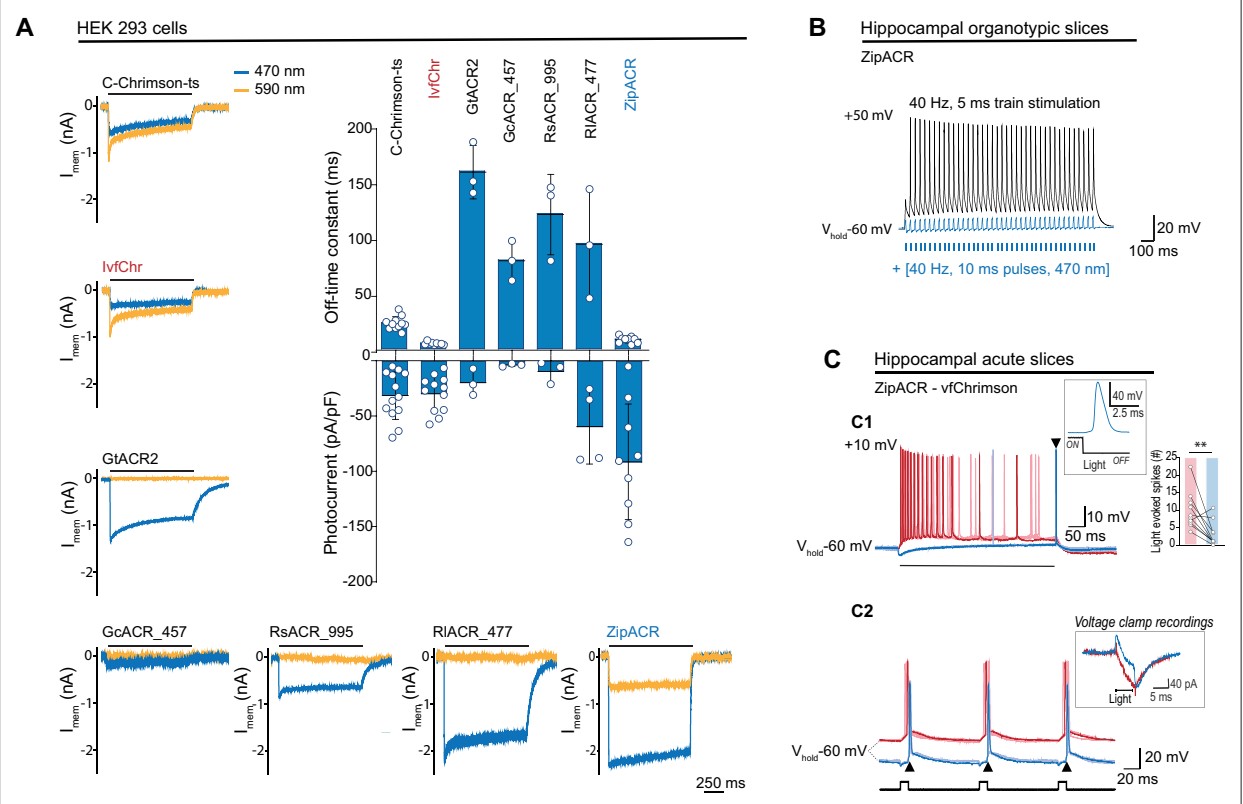

**Figure 2.** Comparison of channel properties of light-gated chloride channels and Chrimson variants. (**A**) Representative photocurrents of the Chrimson and ACRs variants in whole-cell patch-clamp recordings from HEK 293 cells with 590 and 470 nm illumination (top black bars, 10 mW/mm²). The plot shows the off-time constant (top) following 1 s of 470 nm illumination, and the peak photocurrent (bottom) at –60 mV (mean ± SD, n = 3–13 cells). (**B**) Typical response of CA1 cells from hippocampal organotypic slices expressing ZipACR to 40 Hz, 5 ms pulses (top black trace) and to an illumination protocol using 40 Hz of 10 ms light pulses at 470 nm (4 mW/mm²) (bottom blue trace). (**C**) Recordings from dentate gyrus granule cells in acute slices expressing ZipACR and vfChrimson. (**C1**) Representative responses to 500-ms illumination (bottom black bar) at 470 and 635 nm (10 mW/mm²) (blue and red traces, respectively). Note, 635 nm illumination evoked action potential (AP) firing while 470 nm illumination only evoked a single AP time-locked to the end of the light pulse (arrowhead). Insets, example of an AP upon 470 nm light termination and plot showing the number of light-evoked spikes at 635 nm (red bar) and 470 nm (blue bar) illumination. n = 10 cells, paired t-test **p < 0.01. (**C2**) Example of light-induced firings by 635 and 470 nm illumination (red and blue traces, respectively, 5 ms pulses (black squares) at 10 Hz). The arrowheads show consistent APs time-locked to the end of the blue light pulse. Inset, example of a voltage-clamp recording of currents evoked by 5 ms pulses of 635 and 470 nm light (red and blue traces, respectively). Light-induced currents with 470 nm illumination were initially outward but turned inward gradually and peaked immediately following the light offset. Note that the late inward current matches the red light-induced current, reflecting a slower kinetics for vfChrimson compare to ZipACR.

The online version of this article includes the following figure supplement(s) for figure 2:

**Figure supplement 1.** Improved membrane trafficking of vfChrimson and ChR co-expression designs.

**Figure supplement 2.** Membrane trafficking of the improved ultrafast vf-Chrimson, IvfChrimson (IvfChr).

## Identifying a suitable pair of ultrafast light-gated anion channel and a red-shifted Chrimson variant

Although GtACR2 proved to be effective in protocols with pulses of long duration, its slow current decay (*Figure 2A*, see GtACR2, 161 ± 24 ms, n = 3) limits its application to optical stimulations of lower frequencies (*Govorunova et al., 2015*; *Govorunova et al., 2017*). For this reason, we searched for a pair of blue-shifted anion channelrhodopsin and red-shifted cation channelrhodopsin with fast and matching kinetics. For a fast R-ChR, we evaluated the recently published vfChrimson, the fastest red-shifted channelrhodopsin available at this time, which can induce APs up to the intrinsic limit of the neurons (*Mager et al., 2018*).

Previously, we found that Chrimson and its variants traffic poorly to the membrane and developed a strategy to improve the trafficking (C-Chrimson-ts, *Bonaventura et al., 2019*). We adapted this modification to improve the trafficking of vfChrimson (referred to as IvfChr, *Figure 2—figure*

*supplement 1*). In our measurements in HEK 293 cells, IvfChr has a channel off-rate consistent with previously reported (*Mager et al., 2018*) (off-rate time constant of 5.6 ± 0.3 ms, *n* = 10 compared to C-Chrimson-ts 21.5 ± 1.2 ms, *n* = 10; mean ± SEM, *Figure 2*, *Figure 2—figure supplement 2*). When comparing C-ChrimsonR-ts and IvfChr photocurrents, we observe 40% reduction of photocurrent of IvfChr (0.086 ± 0.055 pA/pF/A.U., *n* = 25) compared to C-ChrimsonR-ts 0.145 ± 0.126 pA/pF/A.U., *n* = 25; mean ± SD, *Figure 2—figure supplement 2B* when the peak photocurrent amplitude was adjusted to membrane capacitance and membrane fluorescence.

To counteract the IvfChr excitation, we estimated that desired properties of the blue light-activated anion ChR should have the off-rate time constant of ~15–30 ms, minimal response to light of wavelength >590 nm and a strong photocurrent. We identified ZipACR, RsACR_995, GcACR_457, RlACR_477, and GtACR2 as possible templates for further development based on published data (*Govorunova et al., 2017*). GtACR2, RsACR_995, and RlACR_477 are spectrally ideal with negligible response to 590 nm light (*Figure 3—figure supplement 1A*), but their kinetics are slow (mean off-rate of 161.0 ± 24.1, 122.7 ± 36.3, and 95.8 ± 49.4 ms, respectively, *n* = 3, *Figure 2A*) which could have undesired long lasting inhibitory effects after light termination. GcACR_457 has very small photocurrents and cannot be characterized accurately, possibly due to poor membrane trafficking. ZipACR, although spectrally not ideal (~515 nm; *Govorunova et al., 2017*) has a kinetic close to the desired range (9.5 ± 3.0 ms, *n* = 9) and it has strong photocurrents (*Figure 2A*).

To test the inhibitory property of ZipACR, we injected 40 Hz somatic current in organotypic hippocampal neurons expressing the channel. Consistent with the original report, individual pulses of 470 nm light resulted in the time-locked suppression of APs (*Figure 2B*). Next, we examined whether ZipACR can block blue light-mediated APs in neurons expressing vfChrimson. Recordings from the granule cells in acute hippocampal slices co-expressing ZipACR and vfChrimson showed that, in 8 out of ten cases 470 nm light induced significantly fewer APs than 635 nm light (*Figure 2C1*), in line with our prediction. However, we consistently observed the appearance of individual APs time-locked to the blue light offset (*Figure 2C1, C2*). We concluded that due to the fast-closing rate, the photoinhibitory current generated by ZipACR decays too early to suppress the remaining excitatory response of vfChrimson. Therefore, we decided to generate variants of ZipACR with slightly slower off-kinetics.

## Optimizing ZipACR off-kinetics

It is technically easier to generate mutations with slower kinetics and blue-shifted spectra than generating variants with faster on- and off-kinetics. Thus, we decided to use ZipACR as a template for further mutagenesis instead of further modifying GtACR2. To identify relevant mutation sites in ZipACR, we generated a homology model of ZipACR with the crystal structure of GtACR1 (*Govorunova et al., 2017*) and identified the putative residues forming the retinal binding pocket in ZipACR (*Figure 3A*). Multiple alignments were performed with the kinetically slower GtACR1 and GtACR2 to identify possible replacement of these residues that would result in slower kinetics and/or small blue-shift in ZipACR without disrupting the channel function (*Figure 3A2*). We tested these variants in HEK 293 cells for photocurrent ratio 590/470 nm LED stimulating light and measured the channel off-rate kinetics and photocurrent amplitude (*Figure 3B*).

Of these candidates, we identified I151V and I151T at the fourth transmembrane domain of ZipACR that resulted in a small increase in channel off-rate time constant (15.5 ± 8.4 ms for I151V and 20.2 ± 6.4 ms for I151T, mean ± SD, *Figure 3B*) compared to the original ZipACR (9.5 ± 2.9 ms, mean ± SD). We also generated double and triple mutants combining the I151V or I151T with Y170F, Y177F, or V229L mutations but most of these double or triple mutants appeared to have smaller photocurrents and off-rate kinetics that could not be fitted well with single exponential back to zero baseline. Based on these results, we concluded the variants I151V (Zip(151V)) and I151T (Zip(151T)) are the best candidates for the pairing and proceeded for further characterizations. Neither Zip(151V) nor Zip(151T) mutations changed the ion permeability of ZipACR as measured by channel reversal potential in potassium gluconate and cesium chloride based intracellular recording solutions (*Figure 3—figure supplement 1B, C*). In terms of light sensitivity, it is desirable that the ZipACR variant would respond to 470 nm light faster and stronger and to red-shifted light slower and weaker than IvfChr at the same intensity (*Figure 3—figure supplement 2*). With both Zip(151V) and Zip(151T), the channel on-rate kinetics have the desirable properties as described above to 470 and 590 nm light, although the Zip(151T) on-rate starts to approach the on-rate of IvfChr at light intensities >5 mW/mm$^2$ at 590 nm

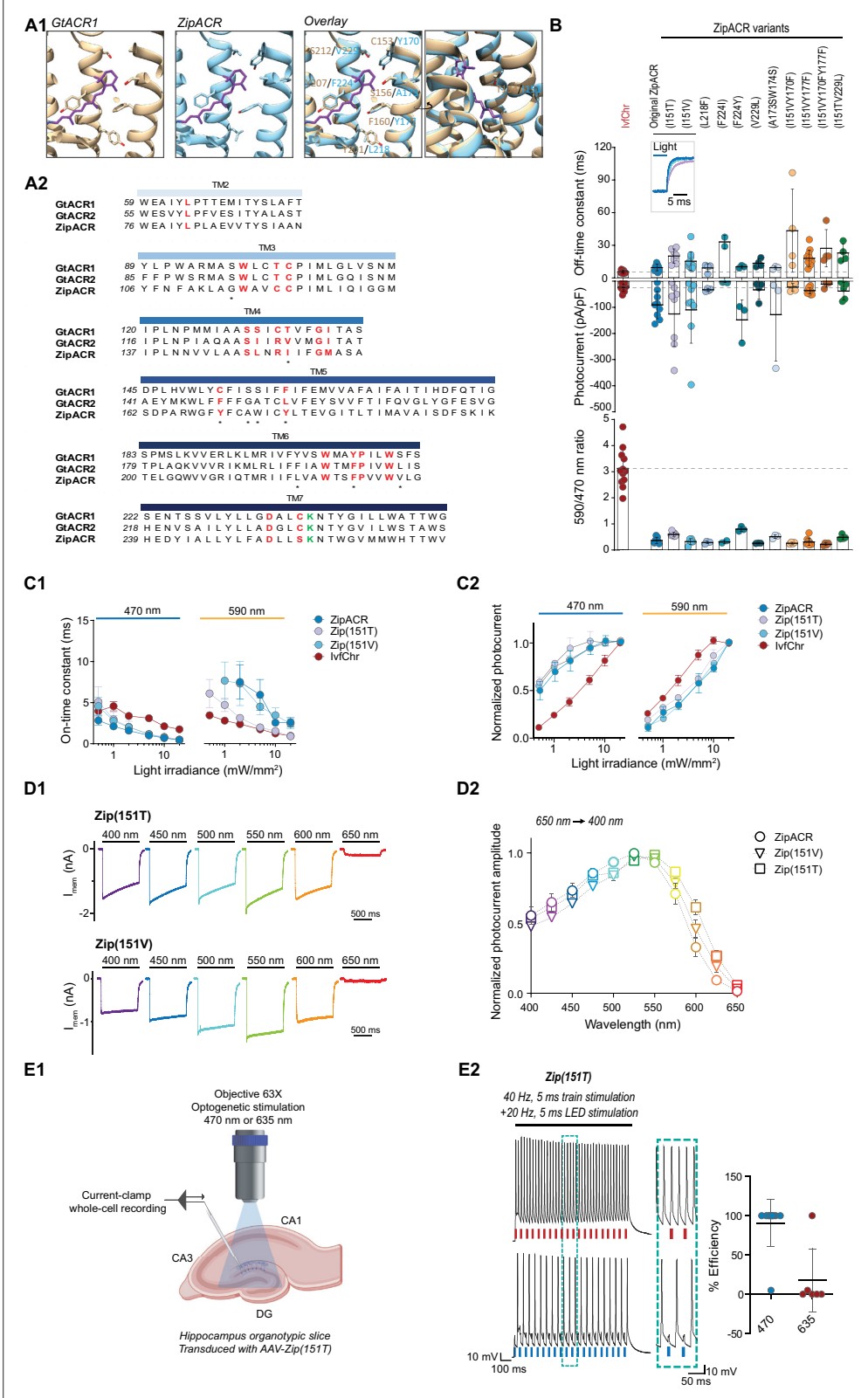

**Figure 3.** Characterization of the optimized variants of the ultrafast anion channelrhodopsin ZipACR. (**A**) Structure-based design of ZipACR variants. (**A1**) The retinal binding pocket of GtACR1 and the putative retinal binding pocket of ZipACR homology model. (**A2**) The alignment of the transmembrane domains (TM) of GtACR1, GtACR2 and ZipACR. The residues surrounding the retinal binding pocket of GtACR1 and the corresponding residues from

*Figure 3 continued on next page*

*Figure 3 continued*

GtACR2 and ZipACR are highlighted in red. *marks the residues targeted for mutation. The lysine that forms the Schiff base with retinal is indicated in green. The numbers on the left correspond to the number of the first residue in each line. (**B**) Basic properties of the ZipACR variants compared to IvfChr. Off-time constant (top) following 1 s of 470 nm illumination, peak photocurrent (middle) and 590/470 ratio (bottom). The data have been obtained from whole-cell recordings in HEK 293 cells. Each dot corresponds to one cell (mean ± SD, *n* = 2–12). (**C**) On-time constant (**C1**) and normalized photocurrent (**C2**) of the selected variants Zip(151T) and Zip(151V) as compared to the original ZipACR and IvfChr, with 1 s 470 or 590 nm illumination of various light intensities (in mW/mm²). For the photocurrent, the values have been measured at the maximum response. For C1, the responses of ZipACR and Zip(151V) to low intensities of 590 nm light were too small for the accurate fitting of the on-time constant and were not recorded. The data points are the mean value ± SEM (*n* = 6–7). (**D**) The responses of the selected variants to various wavelengths of the same photon flux. (**D1**) The action spectra of ZipACR, Zip(151V) and Zip(151T) as measured from 650 to 400 nm (**D2**) For the action spectra, the maximum photocurrent amplitudes measured at each wavelength were normalized to the peak values obtained from the same cell across the spectrum. The data points are the mean value ± SEM (*n* = 6–7). (**E**) Validation of the efficiency of the Zip(151T) variant in slices. (**E1**) The response of dentate gyrus granule cells expressing Zip(151T) variant at 470 nm (4 mW/mm²) and 635 nm (7 mW/mm²) light pulses delivered through a 63× objective. Created with BioRender. (**E2**) Representative traces of the firing induced by current injection at 40 Hz. Overlapping 470 nm but not 635 nm pulses block action potentials. The plot on the right shows the efficiency (in %) of Zip(151T) in blocking individual action potentials at 470 and 635 nm light pulses (mean ± SD, *n* = 6–10 cells). The traces of the two outliers are presented in *Figure 3—figure supplement 3A*.

The online version of this article includes the following figure supplement(s) for figure 3:

**Figure supplement 1.** Channel properties of the ZipACR variants in HEK 293 cells.

**Figure supplement 2.** Light-induced responses and action spectra of ZipACR variants.

**Figure supplement 3.** Zip(151T) fails to produce AP under 470 nm illumination.

---

(*Figure 3C1*). With 470 nm light, IvfChr does not appear to reach saturating response even at 20 mW/mm² (*Figure 3C2*). With 590 nm light, IvfChr reaches saturating response at 10 mW/mm² whereas none of the ZipACR variants appears to reach saturating response even at 20 mW/mm² (*Figure 3C2*). Spectrally, Zip(151T) shows a wider red-shifted action spectra than ZipACR whereas Zip(151V) may have a slightly narrower blue-shifted action spectra than ZipACR (*Figure 3D*).

Since Zip(151T) is a more red-shifted variant of the two (*Figure 3B, D*), we asked whether red light pulses may induce photoinhibitory currents that suppress APs. We expressed Zip(151T) in the granule cells of the dentate gyrus in hippocampal slices using AAV vector and tested its inhibitory effect upon 470 and 635 nm light pulses. We injected somatic current at 40 Hz, while alternately delivering overlapping 5 ms pulses of light. Except in two instances, blue light pulses were consistently effective in blocking APs, while red light had no effect on APs induction (*Figure 3E*, *n* = 10 for 470 and *n* = 6 for 635, the two outliers are presented in *Figure 3—figure supplement 3A*). It has been reported that at resting or hyperpolarized membrane potentials ZipACR may produce light-evoked APs (*Kato et al., 2018*). We did not observe such a phenomenon under our experimental setup (*Figure 3—figure supplement 3B*).

## Validation of the co-expression strategy in brain slices

Since co-injection of two viruses results in variability in the ratio of expression in different neurons, we used the bicistronic 2A approach (Supplementary material 1). We favored this system to the internal ribosomal entry sites (IRES) (*Pelletier and Sonenberg, 1988*; *Douin et al., 2004*) and tandem fusion strategies (*Kleinlogel et al., 2011*). As observed by us and reported by other groups, IRES produces an unbalanced expression level with the second protein at much lower level (*Mizuguchi et al., 2000*). The testing of vfChrimson in the recently published BiPOLES tandem fusion system with GtACR2 (*Vierock et al., 2021*) resulted in high levels of intracellular accumulation of the tandem opsins in HEK 293 cells (*Figure 2—figure supplement 1*).

Therefore, we generated constructs containing IvfChr and Zip(151T/V) (ZipT-IvfChr and ZipV-IvfChr) using the bicistronic 2A cassette with KV2.1 soma targeting sequence for both ZipACR mutant and IvfChr (*Figure 4*, *Figure 2—figure supplement 1*, and Supplementary material 1). This ensures an equimolar translation of the two proteins from one mRNA and soma enrichment of the opsins in neurons. In comparison to BiPOLES with vfChrimson, the vfChrimson in our bicistronic 2A cassette led

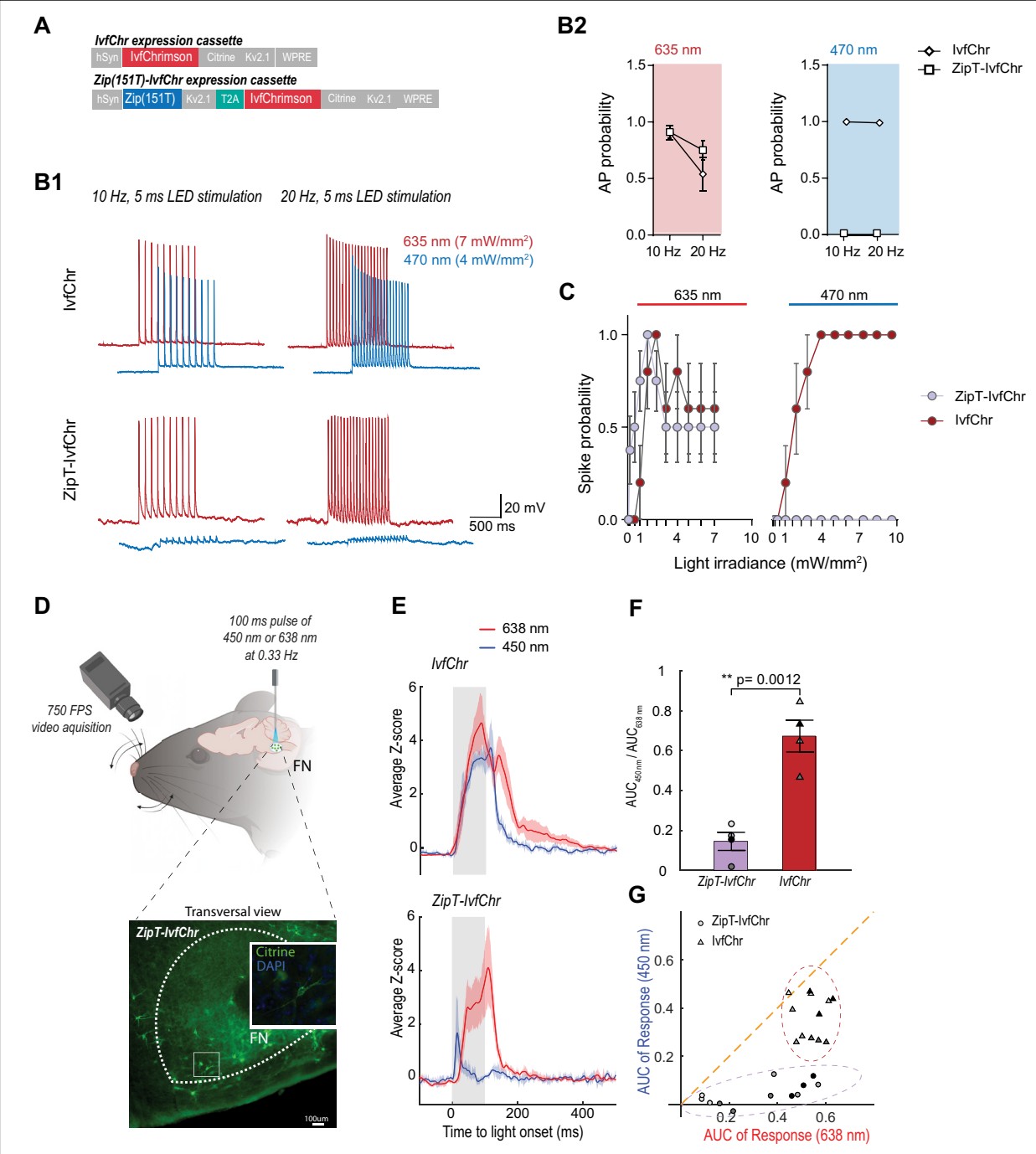

**Figure 4.** Zip(151T) variant co-expressed with IvfChr prevents blue light-induced action potentials, in vitro and in vivo. (**A**) The expression cassette for IvfChr and bicistronic ZipT-IvfChr used in rAAV vector. The selected ZipACR variant Zip(151T) is co-expressed with the red-sensitive IvfChr by using the 2A self-cleaving peptide (T2A). All the opsins are soma-targeted with a Kv2.1 peptide. (**B**) Comparison of the high-frequency red and blue light-induced spiking (**B1**), and spike probability (**B2**), $n$ = 6–13 for each opsin, of DG cells in organotypic slices expressing IvfChr or ZipT-IvfChr, with 10 and 20 Hz of light pulses (4–7 mW/mm²). Holding potential = –60 mV. (**C**) Blue and red light-driven spike fidelity at various light intensities in DG cells ($n$ = 4–10 cells for each opsin; 10 ms pulse width, single pulse). (**D**) Experimental design. ZipT-IvfChr or IvfChr are expressed in the facial nucleus (FN) of the brainstem (top) and light-induced whisker movements are recorded. Example of soma-targeted expression of our ZipT-IvfChr construct in FN (bottom). Created with BioRender. (**E**) Averaged Z-scores of the angle of the whiskers under 450 and 638 nm illumination (20 mW/ mm²) of the FN motoneurons. As opposed to IvfChr, ZipT-IvfChr failed to reliably elicit whiskers movement under blue light ($n$ = 4 animals per condition, see also Supplementary material 2). (**F**) Ratio of the area under the curve (AUC) of the responses to 450 and 638 nm illumination for ZipT-IvfChr and IvfChr ($n$ = 4 animals per condition, **p < 0.01, $t$-test). (**G**) Distribution of the AUC for three individual trials from each animal at 450 and 638 nm illumination ($n$ = 4 animals per condition).

*Figure 4 continued on next page*

*Figure 4 continued*

The online version of this article includes the following figure supplement(s) for figure 4:

**Figure supplement 1.** ZipV-IvfChr variant, as for ZipT-IvfChr, is effective in preventing blue light-induced action potentials (APs) in vitro.

**Figure supplement 2.** Desensitization of IvfChr in HEK 293 cells.

to ~2.7× higher photocurrent (–7.57 ± 5.37 pA/pF in ZipV-IvfChr compared to –2.72 ± 1.33 pA/pF in BiPOLES, *n* = 14) when tested in identical conditions, consistent with the improved membrane trafficking observed with IvfChr (*Figure 2—figure supplement 1A, B*). The photocurrent amplitudes of the IvfChr and ZipV-IvfChr constructs in HEK 293 cells are comparable after adjustment to cell size and membrane fluorescence (*Figure 2—figure supplement 1D*). We did not test the ZipT-IvfChr variant as the red-shifted response of Zip(151T) makes it more difficult to isolate the IvfChr response to 590 nm light available on our LED system.

After characterization in HEK 293 cells, we examined the power at which responses to 470 and 635 nm lights induce APs in neurons expressing ZipT-IvfChr, ZipV-IvfChr, or IvfChr. We expressed AAV constructs in the granule cells of the dentate gyrus in the hippocampal slices (as described in *Figure 3E1*). We recorded cells under the current-clamp mode and injected current through the patch pipette to maintain the membrane potentials at −60 mV. To examine the efficiency of 635 and 470 nm lights in inducing APs, we delivered 10 and 20 Hz light pulses of 5 ms duration (*Figure 4B*). As expected, in IvfChr-expressing neurons, both 470 and 635 nm lights induced APs (*n* = 6). In neurons expressing ZipT-IvfChr (*Figure 4B*), 635 nm pulses were equally effective in driving APs. On the other hand, 470 nm light only induced modest depolarization, which did not produce the threshold for AP induction (*n* = 5–12 for each opsin). Similarly, co-illumination with 470 and 635 nm also failed to generate AP, showing blue light effectively nullifies the depolarizing property of red light (*Figure 4—figure supplement 1A*). In IvfChr-expressing neurons 470 and 635 nm light-induced APs with light powers as low as 1 mW/mm$^2$ (*n* = 10, *Figure 4C*). In contrast, ZipT-IvfChr-expressing neurons produced APs only to 635 nm light stimulation (*Figure 4—figure supplement 1B*), whereas 470 nm light powers as high as 10 mW/mm$^2$ failed to induce APs (*n* = 7–8, *Figure 4C*). As for ZipV-IvfChr, we occasionally observed blue light-induced APs (*Figure 4—figure supplement 1B*), which may be explained by its relatively faster closure rate (*Figure 3B*). In neurons expressing IvfChr, we noticed a reduction in probability of AP upon successive optical stimulation specifically to 635 nm pulses (*Figure 4B*). Consistent with this, we observed a reduction in response in HEK 293 cells expressing IvfChr (*Figure 4—figure supplement 2*). This phenomenon, likely caused by the desensitization of the channel, is consistent with previous reports with other red-shifted channelrhodopsins (*Lin et al., 2013*).

## Validation of the co-expression strategy in vivo

To evaluate the system in vivo, we compared the performance of ZipT-IvfChr and IvfChr in driving vibrissae movements by optically activating motoneurons of the facial nucleus (FN) expressing either of the two constructs (*Lin et al., 2013*; *Sreenivasan et al., 2015*; *Gharaei et al., 2020*; *Figure 4D*). Four to six weeks after virus injection, a 200-μm diameter optic fiber was placed above the FN of head-fixed anesthetized mice and motoneurons were stimulated at 0.33 Hz using 100 ms pulses of either 450 or 638 nm light. In mice expressing IvfChr, whiskers movements of large amplitude were elicited with both red and blue pulses (*Figure 4E–G*, the area under the curve (AUC) of 2.9 ± 0.09 and 1.95 ± 0.13, respectively, with a ratio AUC$_{450}$/AUC$_{638}$ = 0.68 ± 0.05, *n* = 4) (Supplementary material 2). Pulses of red light elicited strong whisking (AUC of 1.8 ± 0.29, *n* = 4) in mice expressing ZipT-IvfChr, which was comparable to the IvfChr group. On the other hand, 450 nm light pulses failed to reliably elicit whiskers movement, as represented by a significant decrease of the AUC under blue light illumination (0.25 ± 0.07, ratio AUC$_{450}$/AUC$_{638}$ = 0.15 ± 0.04, *n* = 4, p = 0.0012, *t*-test) (Supplementary material 2). Overall, we confirmed that in vivo, unlike motoneurons expressing IvfChr, motoneurons expressing ZipT-IvfChr are largely non-responsive to blue light, while preserving the red-shifted property of an R-ChR.

## Fast recovery of blue light-induced suppression of APs invitro

We expected that blue light pulses in ZipV-IvfChr- and ZipT-IvfChr-expressing cells would have an overall inhibitory effect on membrane potential due to the activation of ZipV/T but this effect is short in duration

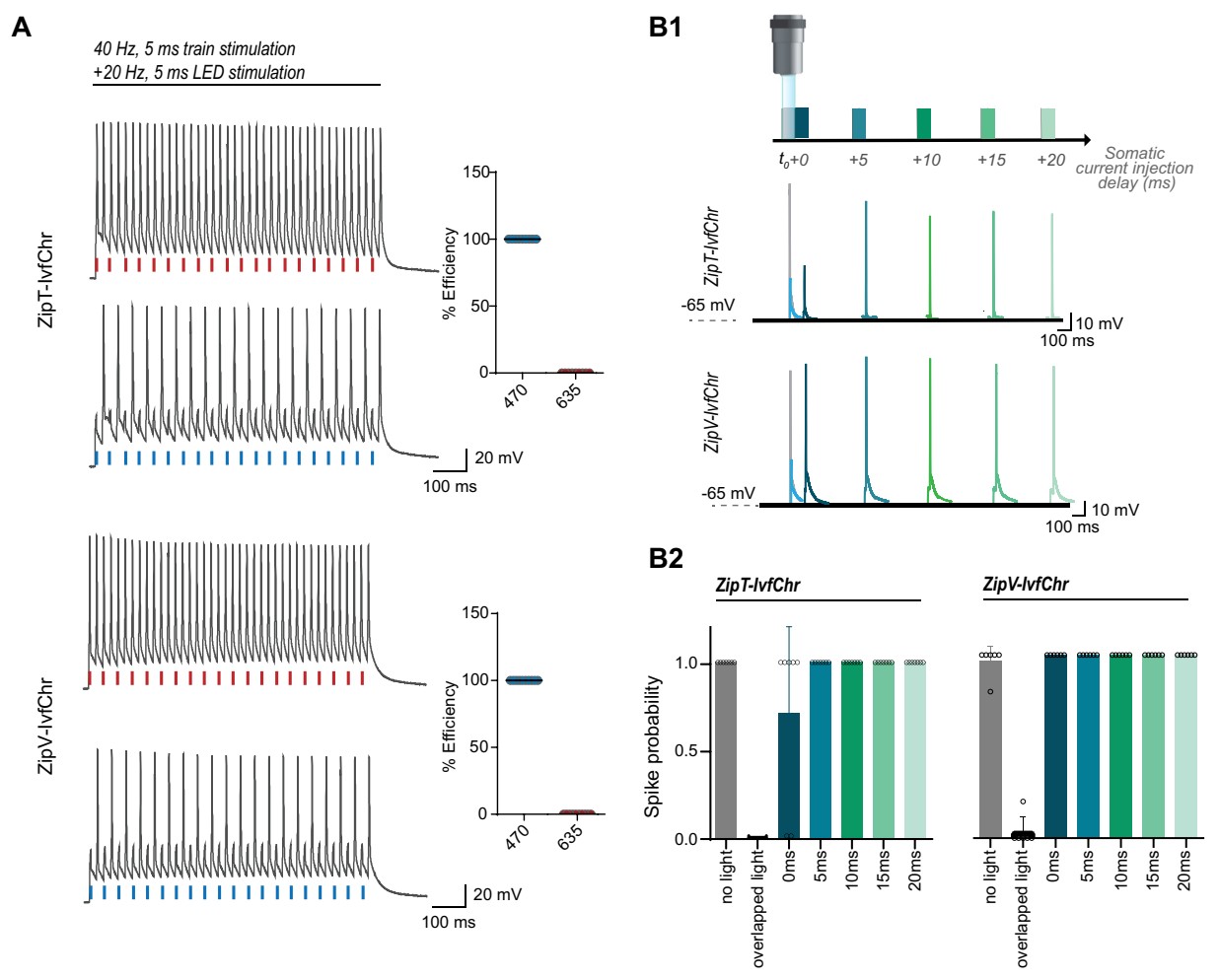

**Figure 5.** The blue light-induced suppression of action potentials is confined within the duration of the light pulse. (**A**) Optical inhibition of spiking in DG cells expressing the ZipT-IvfChr and ZipV-IvfChr variants, in organotypic slices. Representative traces of the firing induced by current injection at 40 Hz. Overlapping 470 nm but not 635 nm light pulses block action potentials. Holding potential = –60 mV. Plots on the right shows the efficiency (in %) of ZipT-IvfChr in blocking individual action potentials at 470 and 635 nm (*n* = 9–10). (**B**) Temporal recovery of neuronal excitability following light-induced inhibition by the ZipACR variants. (**B1**) Experimental paradigm. DG granule cells expressing ZipT-IvfChr or ZipV-IvfChr have been stimulated by a somatic current injection (5 ms pulse) overlapping or preceded by a 470-nm pulse (5 ms, 4 mW/mm²) at 0, 5, 10, 15, or 20 ms intervals. The traces in current clamp (bottom) are representative of the responses to current injections. (**B2**) The spike probability shows a fast recovery of ZipT-IvfChr (<5 ms) and an immediate recovery of ZipV-IvfChr, from the off-set of the 470 nm light (*n* = 7 for ZipT-IvfChr, *n* = 6 for ZipV-IvfChr, mean ± SD).

The online version of this article includes the following figure supplement(s) for figure 5:

**Figure supplement 1.** Evaluation of light-gated potassium channels HcKCR1, HcKCR2, and WiChR.

due to the fast kinetics of ZipACR variants. To test this, we stimulated the neurons by somatic current injection at 40 Hz, while alternately delivering perfectly overlapping 5 ms pulses of lights (*Figure 5A*). Light pulses of 470 nm blocked APs, while 635 nm light pulses had virtually no effect. Importantly, we asked how fast a neuron recovers from the blue light inhibition associated with the expression of ZipV-IvfChr and ZipT-IvfChr. We induced APs through somatic current injection at different time points from the offset of a light pulse (*Figure 5B1*). In neurons expressing ZipT-IvfChr, current injection evokes APs with high probability after 5 ms, which is consistent with our goal of minimizing the unintended disruption of membrane excitability after the light stimulation (*Figure 5B*). The recovery time for ZipV-IvfChr was immediate, with 100% success rate immediately following the light offset (*Figure 5B2*).

## Evaluation and modification of recently developed light-gated potassium channelrhodopsins

A main drawback of using chloride channels is that their opening in the axonal terminals causes depolarization and neurotransmitter release (*Turecek and Trussell, 2001*; *Mahn et al., 2016*). This is due to the high concentration of chloride in the terminals. Since potassium channels are hyperpolarizing in all neuronal compartments, we evaluated the recently published potassium-selective channelrhodopsins HcKCR1, HcKCR2 (*Govorunova et al., 2022*), WiChR, and B1ChR2 (*Vierock et al., 2022*) for possible pairing with IvfChrimson in next version of our tool. We observed significant light-induced membrane currents from HEK 293 cells in all the examined potassium-selective channelrhodopsins, except for B1ChR2 (*Figure 5—figure supplement 1A*). This is consistent with previously reported data in mammalian cells (*Vierock et al., 2022*). Therefore, B1ChR2 was not tested any further.

The channel reversal potentials, as measured by a ramp protocol, were –58.0 ± 8.7 mV ($n = 9$), –50.7 ± 8.0 mV ($n = 14$), and –65.2 ± 6.5 mV ($n = 7$) for HcKCR1, HcKCR2, and WiChR, respectively (*Figure 5—figure supplement 1B, C*) which were more hyperpolarized than the reversal potential for ZipACR measured previously in similar conditions (–39.0 mV). In terms of the channel kinetics, HcKCR1, HcKCR2, and WiChR have the off-rate time constants (measured at holding potential of –10 mV) of 49.7 ± 11.3 ms ($n = 9$), 58.6 ± 47.8 ms ($n = 14$), and 229.3 ± 53.8 ms ($n = 9$) (*Figure 5—figure supplement 1D*) which is much slower than the channel kinetics of ZipACR.

As an observation, the off-rate for the HcKCRs often do not fit a single exponential function well and the value measured here underestimates the speed of the channel (see the traces in *Figure 5—figure supplement 1E1*). HcKCR1 has a spectral peak at 530 nm as previously reported (*Govorunova et al., 2022*) and still responds strongly to 590 nm light (73.0 ± 6.4% of 530 nm response, $n = 10$) whereas HcKCR2 and WiChR have spectral peaks at 470 nm and have small responses to 590 nm light (9.0 ± 3.0% of 470 nm response, $n = 8$ for HcKCR2 and 27.6 ± 7.2% of 470 nm response, $n = 6$; for WiChR; *Figure 5—figure supplement 1E2*). In light of these results, none of the wild-type K-ChRs appear to be optimal for the pairing and further mutagenesis, and engineering in future study would be required to optimize the pairing.

## Discussion

Here, we describe a dual-color expression system by which pulses of red-light drive time-locked high-frequency APs, while blue light allows precise suppression of AP confined within the duration of the pulse. We created this system by pairing a B-ACR with the ultrafast red-shifted ChR, IvfChr. This requires the B-ACR to have a significantly larger photocurrent compared to R-ChR as well as a faster channel opening and slightly slower channel closing. As we discuss later, we considered anion channels with slow decay of inhibition are not suited for many experimental manipulations of long duration. For this reason, we chose ZipACR, currently the fastest B-ACR, as the template for further modifications. Based on our homology modeling and multiple alignments with the kinetically slower B-ACRs, GtACR1, and GtACR2, we identified potential residue replacements within the putative retinal binding pocket which would result in slightly slower kinetics without impairing the channel function.

To our surprise, many mutations of ZipACR around the all-trans retinal do not alter the channel properties or compromise channel conductance very strongly. This is contrary to our past experiences with channelrhodopsins where conductance is highly sensitive to mutations around the retinal binding pocket (data not shown). ZipACR double mutants appear to have more compromised conductance or altered properties. Among the tested positions, the I151 residue appears to have significant influences in kinetics and spectral properties. Indeed, we found that two variants, Zip(151T) and Zip(151V), have properties optimal for our purpose. More specifically, the channels off-rate kinetics have the desirable property of being slightly but sufficiently slower than IvfChr without compromising the opening rate, photocurrent and light spectrum. Thus, when paired with IvfChr, they prevent AP induction by blue light, without interfering with red light-induced high frequency neural spiking.

Of the two variants, we observed Zip(151T), and consequently ZipT-IvfChr, to be more consistent in blocking blue light-induced APs, which could be explained by its slightly slower closing rate. Although Zip(151T) is the more red shifted of the two, within the tested range of light intensities and expression, only rarely did we observe a block of APs at 635 nm. However, we do not recommend the use

of orange light pulses with ZipT-IvfChr, as we observed a significant photocurrent in this wavelength (*Figure 3D1, D2*; *Figure 3—figure supplement 2*). Comparing the ZipV-IvfChr and the ZipT-IvfChr constructs, we would recommend users to use the ZipT-IvfChr for most general experiments if fast light source with wavelength >600 nm is available given the reduced probability of neuronal spiking at the end of blue light pulse. The ZipV-IvfChr system is faster in kinetics and is suitable in situation where kinetics is critical but there is some tolerance in spiking at the end of blue light pulse in some neurons.

A recent study (*Vierock et al., 2021*) proposed an R-ChR expression system by relying on the pairing between GtACR2 and Chrimson, as we did in our early stage work with *Drosophila*. For experimental reasons linked to controlling neuronal physiology, the priority we set was to create a system with fast closing kinetics to minimize any physiological disruption after the termination of light illumination. All channelrhodopsins have the potentials to alter osmotic balance, ion gradient and pH of small neuronal compartments and this is exacerbated by long light pulses and prolonged channel opening. This can be problematic in long-term neuroplasticity experiments. Cationic channelrhodopsins are known to conduct calcium ions, long opening can evoke significant elevation of intracellular calcium (*Lin et al., 2013*, *Lin et al., 2009*) and potentially activate second messenger pathways.

The alteration of pH, osmotic balance, and ion gradient (either directly or secondary to elevated transporter activity) applies to both cationic and anionic channelrhodopsins and can have unpredictable biological consequences. Hence, channelrhodopsins with rapid kinetics such as the ones used in our system and brief stimulation light pulses are more ideal for experiments where neuronal or synaptic plasticity are of importance. We were able to demonstrate that neurons expressing ZipT-IvfChr can recover from this inhibition within 5 ms after light termination and ZipV-IvfChr recovers almost instantaneously. If we model our results to a B-ACR with slower kinetics such as GtACR2 used in BiPOLES, we could expect the inhibitory effects of blue light to last up to 10–20 ms. This could disrupt normal neuronal physiology and signal processing that occurs in millisecond time scale, as well as alter intracellular ion concentrations which may lead to short- and long-term alteration in physiology.

From an experimental viewpoint, a fast photostimulation system has the flexibility to perform a wide range of optical manipulations. Ultimately, ZipT-IvfChr and ZipV-IvfChr will allow dual optical activation of segregated groups of neurons when co-expressed with a fast blue-shifted channelrhodopsin such as oChIEF in the same brain region. In these experiments, it is critical to minimize the blue light-mediated inhibition due to possible disruption of responses resulting from local monosynaptic connection that may have important modulatory effects on postsynaptic cells. Spike-timing-dependent experiments requiring high temporal precision would be one of the experiments that would benefit from our system. Blue light-activated cationic channelrhodopsin such as oChIEF can be expressed presynaptically to control neurotransmitter release and ZipV/T-IvfChr can be expressed postsynaptically to control spiking to induce long-term potentiation or long-term depression.

In cortical experiments, blue cationic channelrhodopsin and ZipV/T-IvfChr can be expressed in projection neurons and interneurons to study the effects of excitation/inhibition balance in cortical processing of information or decision making. In the striatum, the independent control of neurons of direct and indirect pathways would benefit from the dual wavelength manipulation. Many of these experiments would require the use of retrograde AAV, cell-type-specific enhancer/promoter or transgenic animals with defined driver line. Although ZipV/T-IvfChr can theoretically be used with single-photon jGCaMP imaging, this would require blue light illumination with duration greater than several milliseconds which will lead to prolonged inhibition of ZipV/T-IvfChr-expressing neurons.

The main limitation of our current approach is that there is limited control of the IvfChr and ZipV/T expression. Although the 2A approach should achieve better translation ratio compared to the use of IRES, the levels of respective proteins would also depend on the amount of protein degradation and the fraction of channelrhodopsins that failed to fold or incorporate the retinal chromophore correctly. This can vary with different neurons in different regions. Therefore, similar to other optogenetic systems, it is important for the users of our system to validate the approach in their cells of interest before commencing behavioral experiments.

There can also be situations where dual virus delivery would be preferable compared to bicistronic expression cassette. As shown in our testing, the tethered-tandem expression system used in BiPOLES reduces the photocurrent of vfChrimson compared to our IvfChr, possibly associated with hindered membrane trafficking. Given the reduced conductance of vfChrimson compared to

Chrimson, a BiPOLES-based tethered-tandem expression system is also not ideal. The development of an improved tunable dual expression system would therefore be beneficial for the neuroscientific community. Ultimately, the co-expression strategy remains a temporary solution to the spectral cross-talk of opsins and the development of a true spectrally narrow channelrhodopsin would still be the preferred solution. However, whether this is achievable due to the chemical properties of the retinal chromophore is questionable.

Before concluding, we must note that our approach, which acts based on chloride flux, is unlikely to be suitable for axonal terminal inactivation, where the intracellular chloride concentration is high (*Mahn et al., 2016*; *Mahn et al., 2018*). Potassium-selective channelrhodopsins, having a hyperpolarized reversal potential in all cellular compartments, may appear better positioned for our purpose. However, as we show in our characterizations, none of the wild-type variants would be optimal for the pairing kinetically. WiChR is kinetically too slow but is the most potent inhibitor. HcKCR1 has strong desensitization, is kinetically too slow and spectrally too red shifted. HcKCR2 is the least potassium selective but is spectrally and kinetically closest to the ideal candidate of the three. Overall, further engineering and modification would still be required to identify the ideal mutants for the pairing similar to what we have done with ZipACR in this study.

At present, for axonal or terminal stimulation, the prior published approaches utilizing differential kinetics and light sensitivity of the opsins or optical manipulation combined with synaptic release properties would still be the preferred options (*Klapoetke et al., 2014*; *Hooks et al., 2015*). Specifically, we refer our readers to a recent work, where we demonstrate effective independent dual optical activation of converging axons, electrophysiologically as well as behaviorally (*Faress et al., 2024*). Thus, the system introduced here complements these studies that are unsuitable for somatic photo-activation. This permits researchers to choose complementary methods based on their experimental needs. Future engineering of a faster and blue-shifted potassium-selective channelrhodopsin would provide and unlock a more comprehensive approach suitable for both somatic and axonal stimulation (*Govorunova et al., 2022*; *Vierock et al., 2022*).

## Materials and methods
### *Drosophila* optogenetic activation experiments
Flies (*Drosophila melanogaster*) were kept on standard food medium (water, cornmeal, oatmeal, sucrose, yeast, agar, acetic acid, and preservative methyl-4-hydroxybenzoate) at 25°C, 60% humidity, and a 12/12-hr light–dark cycle. Male flies were collected after eclosion and aged for 7 days in isolation and protected from light on standard food, followed by 3 days on food containing 400 µM all-trans retinal (Sigma). The following genotypes were assayed: co-expression of Chrimson and GtACR2: w-; NP2631.GAL4/UAS-CsChrimson.mVenus.attp40; tubP-FRT-stop-FRT-GAL80, fruFLP/UAS-GtACR2. attp2; co-expression of Chrimson and tdTomato: w-; NP2631.GAL4/UAS-CsChrimson.mVenus.attp40; tubP-FRT-stop-FRT-GAL80, fruFLP/UAS-myr-tdTomato. attp2; co-expression of GtACR2 and tdTomato: w-; NP2631. GAL4/UAS-myr-tdTomato.attp2; tubP-FRT-stop-FRT-GAL80, fruFLP/UAS-GtACR2. attp2.

For optogenetic activation experiments, we used a previously described multi-channel array of electret condenser microphones (CMP-5247TF-K, CUI Inc), amplified with a custom-made circuit board and digitized with a multifunction data acquisition device (NI USB-6259 MASS Term, National Instruments) (*von Philipsborn et al., 2014*). Single male flies were placed in recording chambers equipped with microphones at the bottom and light diodes (Amber 591 nm: Cree 5 mm C503B-AAS/ AAN-015 or Blue 470 nm: Cree 5 mm C503B-BAS-CY0C0461) were placed 1 cm above the song recording chambers at a 60° angle, illuminating the whole chamber (10 s of constant light). Tentative courtship song was detected by a MATLAB script and corrected manually, using a custom-made user interface for visualization and annotation of song oscillograms (*von Philipsborn et al., 2014*).

### Cell culture, transfection, and stable cell line generation
HEK 293A cells (Invitrogen) were grown in DMEM (1 g/l glucose) supplemented with 8% FBS and 1% Pen/Strep. For recordings, cells were plated at low density on glass coverslips and kept in DMEM media supplemented with 4% FBS and 1% Pen/Strep. The HEK 293 cells were validated by Australian Genome Research Facility (AGRF) based on genomic DNA STR profiling using the GenePrint10

system (Promega). The HEK 293 cells were grown in Mycoplasma-free conditions (tested by PCR). Transfection was done with Xtremegene 9 (Sigma-Aldrich) according to the manufacturer's instructions. Recordings of the cells were performed between 38 and 50 hr after transfection.

For generation of recombinant lentivirus for stable HEK 293 cell lines expressing IvfChr and ChrimsonR-tdTomato cells, HEK 293 cells were co-transfected with psPAX2 and pMD2.G (courtesy of Trono laboratory) and the media was transferred to a new dish with untreated cells 2 and 3 days after transfection. The cells receiving the media were kept as stable cell lines of respective constructs.

## Structure alignment and modeling

Homology modeling of ZipACR structure was performed with Modeller within the UCSF Chimera to the published GtACR1 structure (PDB #6CSM). This was further aligned to the GtACR2 alignment from *Govorunova et al., 2017*. Residues surrounding the retinal binding pocket of GtACR1 and the modeled ZipACR were identified and those residues that were different in ZipACR were changed to the corresponding residues of GtACR2 or GtACR1.

## Molecular biology

For GcACR_457, RsACR_995, and RlACR_477, the cDNA were in the original pcDNA3.1 vector as EYFP fusion as available on Addgene (#103137, 103772, and 103771). The GtACR2 construct was synthesized (IDT DNA) and modified with a N-terminal signaling peptide from the commercial pDisplay vector (Invitrogen) and a trafficking signal prior to tdTomato or citrine fusion and inserted in the pcDNA3.1+vector (Invitrogen) for expression. For the ZipACR constructs, the original EYFP fusion from Addgene (#88844) was retained and the construct was either inserted into the pcDNA3.1+ vector or a second generation lentiviral vector pLenti with CMV promoter. No functional or fluorescent differences in the expressed protein were observed between expression from pcDNA3.1+ and pLenti vector in HEK 293 cells and the results were combined for comparison and analysis.

Site-directed mutagenesis of the ZipACR, Chrimson and GtACR2 were performed with overlap extension PCR using Phusion DNA Polymerase (Life Technologies) and ligated into the pcDNA3.1+ or pLenti expression vector.

For ChrimsonR and vfChrimson, additional improvements for membrane trafficking were used as previously described (*Bonaventura et al., 2019*). In brief, the N-terminal leading sequence from ChR1/oChIEF/ReaChR was transplanted to the transmembrane A–G of ChrimsonR and vfChrimson. A trafficking sequence was added after the transmembrane G before fusion to tdTomato. The modified ChrimsonR/vfChrimson-tdTomato constructs were inserted into the second-generation lentiviral pLenti vector with CMV promoter and used to make stable ChrimsonR/vfChrimson-tdTomato HEK 293 cell lines. The improved vf-Chrimson is named IvfChr.

For the comparison of IvfChr-citrine and Zip(151V)- Kv2.1-2A-IvfChr-citrine-Kv2.1 (ZipV-IvfChr) photocurrents, the two cDNAs were inserted into pcDNA3.1+for expression. The IvfChr in both designs contain the modified N-terminal sequence from ChR1/oChIEF/ReaChR but the trafficking sequence was only used in the IvfChr-citrine design and not the Kv2.1 design.

For the expression of HcKCR1, HcKCR2, and WiChR, the mammalian codon-optimized sequences of the three potassium-selective ChRs were synthesized as gBlocks (IDT DNA) based on the published sequences and ligated into pcDNA3 or pcDNA3.1H(+) vectors 5′in-frame of coding sequences of EYFP or citrine using BamHI and NotI sites. Trafficking sequences were placed between potassium-selective ChRs and fluorescent protein. N-terminal signaling peptide from IvfChr was placed in some constructs tested. Membrane currents were similar in the presence and absence of N-terminal signaling peptides and the results were combined.

ZipT, ZipV, IvfChr, and bicistronic expression cassettes are available in AAV2 vector on Addgene. org under John Lin laboratory (#221614-221619).

## rAAV production

rAAV2-retro containing vfChrimson_EYFP_Kv2.1, IvfChr_Citrine_Kv2.1, Zip(151T)- Kv2.1_IRES_ DsRed, Zip(151T)-Kv2.1-2A-IvChrimson-citrine- Kv2.1 (ZipT-IvfChr), and Zip(151V)-Kv2.1-2A-IvfChr-citrine-Kv2.1 (ZipV-IvfChr) were produced according to the protocol here. In brief, HEK 293A cells (Life Technologies, Carlsbad, CA) were grown to 90% confluence and transfected with vectors containing opsin proteins, and the helper plasmids XX6-80 and rAAV2-retro (gift from A. Y. Karpova, Howard

Hughes Medical Institute Janelia Farm Research Campus). rAAV2-retro were released and purified according to protocol here.

Alternatively, virus vectors were produced at the University of Copenhagen, using triple-transfection protocol and purified with an iodixanol density gradient column following *Challis et al., 2019* protocol. In brief, 420 million HEK 293t cells were seeded in DMEM 1965 (Glutamax, 10% FBS, 1% P/S). The day after, cells were transfected with 400 µg DNA in a 1:4:2 ratio (pAAV:rep/cap:pHelper), using linear PEI polyethylenimine (PEI, MW 25000). Eighteen hours post-transfection, the media was exchanged to low serum DMEM 1965 (Glutamax, 1% FBS, 1% P/S) and incubated for 72 hr before harvesting AAV vector from both media and cells using PEG8000. The AAVs were purified from an iodixanol density gradient column following 15 hr ultracentrifugation (rotor SW28, 28,000 rpm, 4°C). Purity was confirmed with SDS–PAGE, and titer (vg/ml) was determined with Quant-iTPicoGreen dsDNA Assay Kit (Invitrogen).

## Electrophysiological characterization and imaging of HEK 293 cells

Whole-cell patch-clamping was performed with Axopatch 200A patch-clamp amplifier with Digidata 1322A data acquisition board and pCLAMP10 software on an Olympus IX73 inverted microscope. Cells were clamped at –60 mV except in experiments with the ramp protocol and experiments comparing the 590 nm light-induced photocurrents between IvfChr-citrine and ZipV-IvfChr-expressing cells. In the experiment comparing the IvfChr-citrine photocurrent to the 2A configuration, cells were clamped at –40 mV to reduce the influence of ZipACR response with IvfChr photocurrent. Series resistance for the recorded cells were mostly under 15 MΩ and series resistance were compensated up to 70% during recordings where possible. For comparison of IvfChr and vfChrimson photocurrent in ZipV-IvfChr and BiPOLES-vfChrimson, the potassium gluconate-based intracellular solution was used and cells were voltage clamped at –40 mV (close to the reversal potential of gtACR2 and ZipACR) when stimulated with 590 nm light.

The extracellular solution was composed of (in mM): 145 NaCl, 3 KCl, 2 CaCl$_2$, 1 MgCl$_2$, 20 D-glucose and 10 HEPES (pH 7.35). For most experiments, the intracellular solution contained (in mM): 110 CsCl, 15 TEACl, 10 HEPES, 5 Cs$_2$EGTA, 1 MgCl$_2$, and 0.1 mM CaCl$_2$ (pH 7.25). In experiments comparing the 590 nm light-induced photocurrents between IvfChr-citrine and ZipV-IvfChr-expressing cells and some ramp experiments, a K-gluconate-based intracellular solution containing: 115 K-gluconate, 10 K$_2$EGTA, 10 HEPES, 5 NaCl, 10 KCl, 2 MgATP, and 0.1 Tris-GTP (pH 7.25), was used. For potassium-selective ChR recordings, only the K-gluconate-based intracellular solution was used.

For imaging the expression pattern of channelrhodopsin-expressing HEK 293 cells, cells on coverslips were imaged with an Olympus BX51 equipped with a 40× water immersion objective and a Hamamatsu Orca Flash 4.0LT sCMOS camera and X-cite LED110 light source. Images were acquired using Micromanager 1.4 with no gain and no electron multiplication and exposure time of 250 ms. Citrine and EYFP expression were imaged with a GFP filter cube (Semrock, FF02-470/30 excitation, FF01-520/35 emission and FF495-Di03 dichroic) and tdTomato was imaged with a TRITC filter cube Semrock, FF01-543/22 excitation, FF01-593/40 emission and FF562-Di03 dichroic. For co-relating photocurrent amplitude and membrane expression, pre-recording images of the cells were captured with an Olympus IX73 microscope with a 40× air objective equipped with Photometrics Cascade II 512 EMCCD camera and excitation by the Thorlabs 4 channel LED (LED4D118 with 405, 470, 530, and 590 nm channels and only 470 and 530 nm light were used for imaging with LED driving currents of 150 and 80 mA, respectively). An identical filter set was used as in the Olympus BX51 microscope. Images were acquired using Micromanager 1.4 with no gain and no electron multiplication and exposure time of 250 ms.

For most channelrhodopsin stimulation experiments 10 mW/mm$^2$ of 470 and 590 nm stimulation lights from Thorlabs 4 channel LED light source (LED4D118) was delivered to the specimen plane. The light intensity was controlled with a Thorlabs DC4104 4 channel controller electronically by the separate analogue outputs of a HEKA ITC-18 DAQ board under the control of WinWCP 5.5 software. In the spectral response experiment, a Sutter VF-5 system with a smart shutter in the neutral density mode was used to achieve cross spectrum stimulation at the same photon flux at 1.586 × 1.015 photons/s/mm$^2$ (25.1 mW/mm$^2$ at 400 nm and 15.4 mW/mm$^2$ at 650 nm) between 400 and 650 nm (25 nm increments). The VF-5 has the following filters and bandwidth settings: 451–398 nm with 15 nm bandwidth, 503–446 nm with 15 nm bandwidth, 564–498 nm with 14 nm bandwidth, 632–557 nm with 14 nm

bandwidth, and 703–627 nm with 13 nm bandwidth. A second Sutter shutter was placed in the light path before the light guide to control the light output. For experiments with ACRs, a 15-s interval was used between stimulation episodes. For ChrimsonR/IvfChr stimulation, stimulation light pulses either have 15-s intervals between stimulation episodes (no preconditioning) or a 15-s interval – 1 s 405 nm light pulse – 15-s interval stimulation pattern for reconditioning of the protein with 405 nm before the subsequent stimulation. The stimulation light was guided into the fluorescent excitation light path of the Olympus IX73 microscope as with imaging but reflected with a Semrock FF685-DiO2 dichroic mirror to the objective.

Light intensity (in mW/mm$^2$) was measured with a Thorlabs slide based light detector (S170C) at the specimen plane with illumination area defined by the microscope aperture and manually calculated for photon flux or illumination intensity.

For measuring the membrane expression of vfChrimson in various modified constructs, BiPOLES and 2A-based cassettes, a membrane localized mCherry (encoded by pcDNA3-mCherry-CaaX) was co-expressed and live cell imaging were done on an Olympus Fluoview FV3000 confocal microscope with 40 × 0.8 NA water immersion objective. The citrine fluorescence was acquired with a 488-nm laser excitation and 500–550 emission. mCherry images were acquired with a 594-nm laser excitation and 600–700 nm emission. For the analysis, single in-focus z-planes where the membrane boundaries were clearly defined were used. A straight line was drawn across the cell between 2 membrane boundaries devoid of the nucleus. The pixel intensities of the citrine and mCherry channels were measured to generate the Pearson correlation coefficient.

## Organotypic slices and virus infection

All procedures involving animals were approved by the Danish Animal Experiment Inspectorate (Permit numbers: 2020-15-0201-00421 and 2023-15-0201-01431) unless otherwise noted. Hippocampi were isolated from postnatal P5-6 wild-type Sprague-Dawley rats (Janvier). The pups were decapitated and the brain removed and kept in an icy low sodium ACSF solution composed (in mM) of 1 $CaCl_2$, 4 KCl, 1 $MgCl_2$, 26 $NaHCO_3$, 10 D-glucose, 234 Sucrose, and 0,1% Phenol Red Solution, in milliQ $H_2O$, and bubbled with a mixture of $CO_2$ (5%) and $O_2$ (95%). From that moment, all the steps were performed in a laminar flow tissue culture hood using sterile equipment and in total asepsis. The brain was poured into a petri dish filled with the low sodium ACSF solution for hippocampus dissection under microscope guidance (Stereomicroscope, Olympus x1 objective). After extraction, the hippocampi were sliced at 400 μm thickness using an automatic tissue chopper (McILWAIN) and moved to a dish with pre-heated culture medium containing MEM Eagle medium 78.8% (Gibco), 20% heat-inactivated horse serum (Gibco), 1 mM L-glutamine, 1 mM $CaCl_2$, 2 mM $MgSO_4$, 170 nM insulin, 0.0012% ascorbic acid, 12.9 mM D-glucose, 5.2 mM $NaHCO_3$, 300 mM HEPES (Sigma), pH = 7.2–7.3, osmolarity adjusted to 315–325. The slices with intact DG and CA regions were then transferred onto air-fluid interface-style Milli-cell culture inserts (Millipore) in 6-well culture plates (Thermo Fisher Scientific) with 800 μl of sterile medium added below each insert. The slices were kept in a sterile incubator at 37°C degrees with 5% $CO_2$ (Thermo Scientific, Steri-cycle i-160). The medium was replaced by a pre-warmed medium (37°C) three times a week.

After 2–3 days of culture, the slices were microinjected in DG (or in CA1 when specified) along the characteristic horseshoe pattern with a pulled glass pipette containing the following viruses: retro-IvfChr (titer: 2.5 × 10$^{12}$ particles/ml), retro-Zip(151T) (titer: 9 × 10$^{12}$ particles/ml), retro-ZipT-IvfChr (titer: 2.9 × 10$^{12}$ particles/ml), or retro-ZipV-IvfChr (titer: 1.5 × 10$^{12}$ particles/ml). The injections were done under a microscope using a Picospitzer III (Parker) connected to a Pulse Pal (10 ms pulse, every 0.5 s). In the process of evaluating the R-ChR expression system in vivo, we noticed variability in the performance between batches from different virus sources. Although we adjusted for the nominal titers given by the providers, some batches produced toxicity while others fared well (data not shown). Therefore, the titers should not be taken as face value, and each batch must be tested in the region of interest before proceeding with the experiment. This concern applicable to any viral expression system has an added complication here. Higher titers, in addition to toxicity, may lead to the leakage of the soma-targeted channels expression in the terminals, which may cause unintended consequences. Recent developments in engineering soma-targeted motifs and peptides must be considered in future works (*Shemesh et al., 2020*).

## Whole-cell electrophysiology and light delivery invitro

Two to three weeks post-infection, the organotypic slices were transferred into the recording chamber and continuously perfused with Artificial Cerebrospinal Fluid containing (in mM): 119 NaCl, 2.5 KCl, 26 $NaHCO_3$, 1 $NaH_2PO_4$, supplemented with 11 D-glucose, 4 $CaCl_2$, 4 $MgCl_2$. Additionally, APV (50 μM), NBQX (10 μM), and picrotoxin (100 μM) were added to block excitatory and inhibitory fast transmission. The solution was adjusted to pH 7.4 (osmolality ~330 mOsm) and bubbled with a mixture of $CO_2$ (5%) and $O_2$ (95%), at room temperature. The chamber was mounted on an upright microscope (Scientifica) linked to a digital camera (QImagingExi Aqua) and the cells were visualized using 60× water-immersion objective (Olympus, LumiPlan). Acquisitions were performed in whole-cell configuration using Clampex 10.6 connected to a Multiclamp 700B amplifier via a Digidata 1550A digitizer (all from Molecular Devices). Voltage-clamp data were low-pass filtered at 200 Hz and digitized at 10 kHz and the whole-cell capacitance was compensated. Patch pipettes (2–4 MΩ of resistance) were filled with an internal solution containing (in mM): 153 K-gluconate, 10 HEPES, 4.5 NaCl, 9 KCl, 0.6 EGTA, 2 MgATP, and 0.3 NaGTP. The pH and osmolarity of the internal solution were close to physiological conditions (pH 7.4, osmolarity 297 mOsm). The access resistance of the cells in our sample was ~25 MOhm. For voltage-clamp experiments, the Dentate Gyrus granule cells (otherwise specified when done in CA1 pyramidal cells) were clamped at –60 mV. For current-clamp experiments, the cells were maintained at –60 mV (except for the protocol where the potential was brought from –70 to –40 mV) by constant current injection (holding current ±100 pA).

The recorded neurons were illuminated with 470 and 635 nm (nominal wavelengths) light with a coolLED pE-4000 system connected to the Digidata via a single TTL. The light irradiance was controlled via the control pod and was adjusted to 4 mW/mm² for 470 nm and 7 mW/mm² for 635 nm (otherwise specified). Light irradiance was measured with a Thorlabs digital optical power meter (PM100D) at the specimen plane with illumination area defined by the microscope aperture. Between the recordings, wavelengths were switched from 470 to 635 nm and vice versa. At the end of the recordings, the location of the recorded cell was confirmed by inspection at ×10 magnification. Data analysis was performed using Clampfit 10.6 (Molecular device).

After recordings, slices were kept in 4% formalin, and later mounted under coverslips with Fluoromount (Sigma-Aldrich). Pictures were taken with an Apotome.2 (Zeiss) and ZEN software at ×10 and ×40 magnifications.

## Acute slices

Experimental procedures were approved by the Animal Care and Use Committee of the University of Buenos Aires (CICUAL 2199/2015). Briefly, 4-week-old C57 mice (n = 4) were co-injected with AAV- 8/2-hSyn1-ChrimsonVF_EYFP_Kv2.1 and AAV-8/2-hSyn1- ZipACR_Kv2.1_IRES_DsRed2 within the dentate gyrus of the hippocampus (250 nl of a 1:1 mixture, coordinates from Bregma: 1 mm mediolateral, –2 mm anteroposterior, –1.5 dorsoventral). After 3 weeks of expression, the animals were sacrificed and the brain removed and cut in a solution composed of (in mM): 234 sucrose, 11 glucose, 26 $NaHCO_3$, 2.5 KCl, 1.25 $NaH_2PO_4$, 10 $MgSO_4$, and 0.5 $CaCl_2$ (equilibrated with 95% $O_2$–5% $CO_2$). The slices were maintained at room temperature before being transferred in a recording chamber mounted on a microscope (Nikon) connected to a Mightex Illumination system for 470 and 635 nm light delivery (at 10 mW/mm²). For current-clamp experiments the amount of current injected was corrected in the inter-sweep interval to keep it close to –60 mV. In voltage clamp experiments holding potential was –60 mV.

Recordings were done in a solution of composition (in mM): 126 NaCl, 26 $NaHCO_3$, 2.5 KCl, 1.25 $NaH_2PO_4$, 2 $MgSO_4$, 2 $CaCl_2$, and 10 glucose (pH 7.4). Patch pipettes (2–4 MΩ of resistance) were filled with a K-gluconate-based internal solution (in mM, 130 K-gluconate, 5 KCl, 10 HEPES, 0.6 EGTA, 2.5 $MgCl_2·6H_2O$, 10 phosphocreatine, 4 ATP-Mg, 0.4 GTP-$Na_3$).

## In vivo expression of ZipT-IvfChr and IvfChr in the FN

For expression of ZipT-IvfChr and IvfChr in the facial motor nucleus of the brainstem (FN), 8- to 10-week-old c57Bl6/jmice (males) from Janvier (France) were anesthetized with Isoflurane (5% for induction, 2% for maintenance) and head-fixed on a stereotaxic frame (Model 940, Kopf, California) on top of a warm plate to keep their body temperature at 37°C. They were administered carprofen (5 mg/kg, s.c.), then the skin was retracted, and a small craniotomy was drilled above the injection

site. Suspension of either retroAAV-hSyn-ZipT-IvfChr (titer: 2.9 × 10$^{12}$ particles/ml) or retroAAV-hSyn-IvfChr (titer: 2.5 × 10$^{12}$ particles/ml) was injected unilaterally at a volume of 400 nl and a rate of 2 nl per second, using a nanoliter injector (Nanoject III, Drummond, Broomal, PA). The coordinates for injection in the FN were –5.6 mm anteroposterior, –1.4 mm mediolateral, and –6.0 mm ventral, relative to bregma. After virus injection, the injection pipette was kept in place for an additional 10 min before being retracted. After they recovered from the surgery on a warm-plate, the animals were put back in their home-cages. They were housed in groups of 4 individuals per cage in a normal 12 hr light/dark cycle with food and water ad libitum. Four to six weeks after the surgery, the animals were anesthetized with Urethane (Sigma-Aldrich, 1.5 mg/kg for first injection, followed by injection of 1/5 of the initial dose until the reflex disappeared) and head-fixed on a KOPF stereotaxic frame (Model 940, Kopf, California). The reflexes were monitored during the entire procedure. The wound on the skin was reopened and the craniotomy above the injection site was re-open when necessary.

For optical stimulation of the FN motoneurons, a 200-μm diameter optic-fiber connected to a laser source (Doric Lenses, Québec, Canada) was implanted above the FN at coordinates –5.6 mm anteroposterior, –1.4 mm mediolateral relative to bregma, and –4.1 to –4.5 mm ventral to the brain surface. Pulses of 100 ms of either 450 or 638 nm illumination at 20 mW/mm$^2$ were given at 0.33 Hz through a PulsePal trigger generator (Open Ephys) connected to the laser. The whiskers' movements were recorded at 750 FPS with an Optronis Camera (CR3000, Optronis GMBH, Germany) and acquired through TimeBench (2.6.30, Optronis). After acquisition, the videos were exported as AVI with a 25 AVI framerate.

At the end of the experiment, the animals were sacrificed and the brain extracted and kept in formalin 10% until slicing. 100 μm sections were sliced with a Vibratom (Leica, VT1200S) and counterstained with DAPI (Thermo Fisher, D1306) and eGFP (Thermo Fisher, CAB4211) for histological localization of the virus expression, and imaged with a ZEISS Apotome microscope (Axio Imager M2). The experimenters were not blind to the group's status; however, the automated analysis did not required blindness as this would not affect the outcome of the experiment.

## Whiskers tracking and analysis

The whiskers were tracked offline with a custom MatLab toolbox (available on GitHub - NabaviLab-Git/WhiskerTrack: WhiskerTrack is a MATLAB toolbox to track the whiskers automatically with high accuracy; *Nabavi Lab, 2022*). In brief, 5–10 whiskers per animal were tracked frame by frame by using an automatic search method to fit the best line to the linear part of each whisker. The slope of this line was considered as the angle of its respective whisker. The absolute changes in the angle of each whisker were compared to its baseline and calculated as a Z-score based on the mean and SD.

The AUC was calculated as AUC = $Z \times W$ with $Z$ being the averaged Z-score for the maximum duration of the response and $W$ being the window of the maximum response (200 ms from the onset of the light pulse).

For comparison between ZipT-IvfChr and IvfChr, we averaged the AUC of the responses to 450 or 635 nm illumination over three trials and calculated the ratio of the AUC to 450 over 635 nm. Statistical significance was calculated with a parametric *t*-test with MatLab.

## Data analysis

The data were analyzed and plotted using GraphPad Prism V9 (GraphPad Software, La Jolla, CA, USA). All values are indicated as mean ± SD or mean ± SEM, as specified. For statistical comparison between groups, data were tested for normal distribution using D'Agostino and Pearson, Shapiro–Wilk or KS normality test. Comparison of the mean between groups has been done with a one-way ANOVA followed by Dunnett's multiple comparisons test.

ZEN software (Zeiss) and FIJI (from ImageJ) were used for images processing.

*Figures 1A, B, 3E, 4D, and 5B1* have been created with BioRender.com.

## Acknowledgements

We thank the members of the Nabavi lab, in particular Islam Faress for suggestions, and Mariam Gamaleldin for the preparation of hippocampi organotypic slices. We also thank Dr. Sergio Almeida for his help on HEK 293 cells culture at the early stage of the project, Peter Kerwin from the Philipsborn lab for his help on *Drosophila* work, and Anne-Katrine Vestergaard, Kathrine Meinecke Christensen, and

Sanaz Ansarifar for technical assistance. We are thankful to Dr. Thomas Mager and Dr. Ernst Bamberg for their generous gift of vfChrimson DNA vector. In the process of writing the manuscript, we greatly benefited from our discussion with Dr. Roberto Malinow. This study was supported by an ERC starting grant to SN (22736), by the Danish Research Institute of Translational Neuroscience to SN (19958), by PROMEMO (Center of Excellence for Proteins in Memory funded by the Danish National Research Foundation) to SN (DNRF133), by the Australian Research Council Future Fellowship (FT160100056) to JYL, NIH BRAIN Initiative grant to JYL (R21EY027620), by a Lundbeck NIH Brain Initiative grant to SN and JYL (R360-2021-650 and R273-2017-179), and by a Lundbeck Foundation grant to Andreas Toft Sørensen (R346-2020-1793 (ATS)).

## Additional information

### Funding

| Funder | Grant reference number | Author |
| --- | --- | --- |
| European Research Council | 22736 | Sadegh Nabavi |
| Danish Research Institute of Translational Neuroscience | 19958 | Sadegh Nabavi |
| Danish National Research Foundation | DNRF133 | Sadegh Nabavi |
| Australian Research Council | FT160100056 | John Yu-luen Lin |
| National Eye Institute | R21EY027620 | John Yu-luen Lin |
| Lundbeck Foundation | R360-2021-650 | John Yu-luen Lin Sadegh Nabavi |
| Lundbeck Foundation | R273-2017-179 | John Yu-luen Lin Sadegh Nabavi |
| Lundbeck Foundation | R346-2020-1793 | Andreas Toft Sørensen |

The funders had no role in study design, data collection, and interpretation, or the decision to submit the work for publication.

### Author contributions

Noëmie Mermet-Joret, Conceptualization, Data curation, Formal analysis, Investigation, Writing – original draft, Writing – review and editing; Andrea Moreno, Formal analysis, Investigation, Methodology, Writing – review and editing; Agnieszka Zbela, Investigation, Methodology; Milad Nazari, Bárður Eyjólfsson Ellendersen, Formal analysis, Investigation; Raquel Comaposada Baro, Nathalie Krauth, Anne von Philipsborn, Andreas Toft Sørensen, Formal analysis, Investigation, Methodology; Joaquin Piriz, Formal analysis, Investigation, Methodology, Writing – original draft, Writing – review and editing; John Yu-luen Lin, Sadegh Nabavi, Conceptualization, Data curation, Formal analysis, Supervision, Funding acquisition, Investigation, Methodology, Writing – original draft, Project administration, Writing – review and editing

### Author ORCIDs

Noëmie Mermet-Joret (ORCID) http://orcid.org/0000-0002-9889-6647
Agnieszka Zbela (ORCID) https://orcid.org/0000-0002-7306-1168
Raquel Comaposada Baro (ORCID) https://orcid.org/0000-0001-8169-1218
Nathalie Krauth (ORCID) https://orcid.org/0000-0001-8048-7119
Anne von Philipsborn (ORCID) https://orcid.org/0000-0002-7921-8744
Joaquin Piriz (ORCID) https://orcid.org/0000-0002-2738-445X
John Yu-luen Lin (ORCID) https://orcid.org/0000-0002-1723-4597
Sadegh Nabavi (ORCID) https://orcid.org/0000-0002-3940-1210

## Ethics

For the animal experiment conducted in Aarhus University, all procedures involving animals were approved by the Danish Animal Experiment Inspectorate (Permit numbers: 2020-15-0201-00421 and 2023-15-0201-01431). The acute slice experiments were approved by the Animal Care and Use Committee of the University of Buenos Aires (CICUAL 2199/2015).

Reviewer #1 (Public review): https://doi.org/10.7554/eLife.90327.3.sa1

Author response https://doi.org/10.7554/eLife.90327.3.sa2

## Additional files

### Supplementary files

Supplementary file 1. The annotated sequence of the 2A-based ZipT-IvfChr design used in this study. For ZipACR(151V), position 151 of ZipACR is valine and encoded by the codon GTC and the sequence of IvfChrimson-citrine starts at polypeptide sequence position 409. Note that the flexible linkers and amino acids introduced by the digestion sites are not annotated.

Supplementary file 2. Area under the curve (AUC) for the whiskers protraction triggered by red or blue light illumination of the facial nucleus neurons expressing ZipT-IvfChr or IvfChrimson.

MDAR checklist

### Data availability

The datasets used and/or analyzed during the current study are available at the following link: https://github.com/NabaviLab-Git/Dual-color-optical-activation-and-suppression-of-neurons-with-high-temporal-precision (copy archived at *Nabavi Lab, 2025*).

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
