## [Editor Report · eLife Assessment]

This study develops **useful** tools for distinct optogenetic control of neuronal activity by red or blue light. The basic characterization of the activation of a red-shifted channelrhodopsin paired with a blue-light sensitive anion channel engineered to obtain desired inhibitory current kinetics is **solid**. However, evidence for their practical use under simultaneous multi-color or high frequency stimulation in cells are missing.

---

## [Referee Report · Reviewer #1 (Public review)]

Summary

In this manuscript, the authors generate an AAV-deliverable tool that generates action potentials in response to red light, but not blue light, when expressed in neurons. To do this, they screen some red light-excitatory/blue light-inhibitory opsin pairs to find ones that are spectrally and temporally matched. They first show that this works with Chrimson and GtACR2, however, they expand their search after finding that the tau-off (inactivation after light cessation) kinetics of these two opsins are not well-matched. They directly examine a small set of options based on a literature search and settle on a variant of red light-excitatory Chrimson and blue light-inhibitory ZipACR. To even more closely match the kinetics of this pair, the authors create a structure homology model of the ZipACR retinal binding pocket and use this to guide generation of a small mutant panel, leading to a more optimized ZipACR mutant. They then show that a bicistronically expressed fusion arrangement of these opsins, plus some functional peptides, can drive action potentials up to 20hz with red light and does not do so with blue light, in hippocampal cells transduced by AAV. They also show function in vivo, in a mouse, using a physiological readout. They conclude that their new tool may be useful for complex experimental designs requiring multiple optical channels for write-in/read-out.

The major advantage claimed by the authors over existing tools is the temporal time-locking of their inhibitory opsin - this is driven by the contrast between tau-off kinetics of their ZipACR variant compared to gtACR2, which is used by the leading competitor tool (BiPOLES).

Big thoughts

While the authors were carefully thoughtful about the potential influence of temporal kinetics on the efficiency of a tool such as this one, there were no experiments conducted that make use of the unique properties of this molecular strategy (although the authors state that these experiments are now underway in their lab). They share some examples of how the tool could be useful in the discussion. Where do I think this could be useful?

First, experimental designs that require multiple optical channels of control. This appears to be aligned with the author's thoughts, as they state, correctly, that opsins utilizing retinal as a light-sensing chromophore are universally activated by blue light (the so-called 'blue shoulder'). Therefore, their tool may be useful for stimulating multiple populations using a blue excitatory opsin in neuron A and their tool for red excitation of neuron B - or, in the author's own words, "A potential solution to the problem of cross-talk...". In this manuscript, the authors provide state that there this is possible in theory and that there are no obvious reasons that it would not work, but do not present data that showcases their new tool for this purpose (e.g. Vierock, Johannes, et al. "BiPOLES is an optogenetic tool developed for bidirectional dual-color control of neurons." Nature communications 12.1 (2021): 4527. Figure 4f-I; 6). The same set-up could be imagined for green GECI (or equivalent) imaging of cells in the same volume that their tool is being used in - for instance, interleaving red stimulation light and blue imaging light, (perhaps) without the typical concern of imaging light bleed-through activating the opsin itself. I agree that it will likely work for multi-channel control, but only time will tell, at this point.

Second, for high-frequency temporal control over both excitation and inhibition in the same neuron. Red light turns the cell on, and blue light turns the cell off (see, for instance, Zhang, Feng, et al. "Multimodal fast optical interrogation of neural circuitry." Nature 446.7136 (2007): 633-639. Figure 2; Vierock as above, Figure 4a,b). Again, here the authors are long on theory ("The new system...can drive time-locked high-frequency action potentials in response to red pulses") and short on explicit data. While they do show that red light = excitation and blue light = inhibition, they neither show (1) all-optical on/off modulation of the same cell; nor (2) high-frequency inhibition or excitation (max stim rate of 20hz, which is the same as the BiPOLES paper used for their LC stimulation paradigm; Vierock, as above, Figure 7a-d). They did provide a response to this critique that data showing excitation and inhibition spread across multiple panels were largely collected from the same cells.

Despite these major shortcomings, the further development and characterization of tandem opsins, such as this one, is of interest to the community. There is on-going work by the BiPOLES team to create new iterations (e.g. Wahid, J., et al. "P-15 BiPOLES2 is a bidirectional optogenetic tool with a narrow activation spectrum and low red-light excitability." Clinical Neurophysiology 148 (2023): e16.). The authors have collected a substantial amount of additional data along the course of review and, even aside from the final tool, the overall data and approaches shown are useful.

---

## [Author Response]

The following is the authors’ response to the original reviews.

**Reviewer #1 (Public Review):**
Therefore, their tool may be useful for stimulating multiple populations using a blue excitatory opsin in neuron A and their tool for red excitation of neuron B… Yet, there are no data presented that showcases their new tool for this purpose

We agree with the reviewer that in this manuscript we have not experimentally shown the applicability of our system for dual optical stimulation. However, the suppression of blue-light excitation of ZipV/T-IvfChr-expressing neurons strongly suggests this can be used in experiments exciting populations of neurons similarly shown for BiPOLES. We don’t see a theoretical basis where this experiment cannot be done if sufficient cell targeting mechanisms (such as the use of cre-lox or retroAAV) is utilized. We have started several projects pursuing these utilities in the meantime.

While they do show that red light = excitation and blue light = inhibition, they neither show (1) all-optical on/off modulation of the same cell; nor (2) high-frequency inhibition or excitation (max stim rate of 20hz, which is the same as the BiPOLES paper used for their LC stimulation paradigm; Vierock, as above, Figure 7a-d).

Regarding point 1, we understand that the reviewer asks if we have optically excited (with red light) and inhibited (with blue light) the same neurons. If so, figure 4B1 (optical excitation of ZipT-IvfCh with red light) and figure 5A (optical inhibition of ZipT-IvfCh with blue light) represent largely the same set of neurons.

Regarding point 2, we respectfully disagree with the reviewer’s interpretation of Figure 7a-d in Vierock et al. As we understand, in this part the authors apply a 20 Hz optical stimulation protocol to the LC neurons in vivo. However, there is no data showing that individual neurons do follow this stimulation protocol. To be clear, we are not saying that BiPOLES cannot drive 20 Hz APs. Very likely it can. It is based on ChrimsonR which is capable of doing so (Klapoetke et al., Figure 2). Although, in this manuscript we have not shown data for optical stimulation above 20Hz, our system is based on vfChrimson, which is known to drive AP of 100Hz and above (Mager et al., figure 2 and 3).

*…* they must revise the manuscript to show that their approach is both (1) different in some way when compared to BiPOLES (it is my understanding that they did not do this, as per the supplementary alignment of the BiPOLES sequence and the sequence of the BiPOLES-like construct that they did test) and (2) that the properties that the investigators specifically tailored their construct to have confer some sort of experimental advantage when compared to the existing standard.

In the latest version of the manuscript, we have compared our ZipV-IvfChr and the BiPOLES construct adapted with vfChrimson (Fig. 2 Suppl 1). The mean photocurrent amplitude of IvfChr in the ZipV-IvfChr construct is ~2.7 x higher than BiPOLES adapted with vfChrimson (14 randomly selected HEK293 cells in each group) (Fig. 2 Suppl 1B). We conducted this experiment in HEK293 cells to ensure accurate voltage-clamping and less biased cell selection. Even adjusting for the smaller photocurrent of vfChrimson vs ChrimsonR, this would still translate to ~1.6 x greater photocurrent with ZipV-IvfChr compared to the original BiPOLES utilizing ChrimsonR. We believe the increased efficiency of excitation is an important aspect of adapting vfChrimson for red-light excitation of neurons.

**Reviewer #2 (Public Review):**
(1) In the Introduction or Discussion, the authors could better motivate the need for a red-shifted actuator that lacks blue crosstalk, by giving some specific examples of how the tool could be productively used, e.g. pairing with another blue-shifted excitatory opsin in a different population, or pairing with a GFP-based fluorescent indicator, e.g. GCaMP. The motivation for the current tool is not obvious to non-experts.

In the discussion, we now provided examples for potential use of the tool. For example, one of the key aspects that can be manipulated by the existing tool is the induction of spike-timing dependent plasticity with 2 wavelengths of light with blue light channelrhodopsin such as oChIEF is used to evoke presynaptic release and ZipT-IvfChr expressed in postsynaptic neuron. In this situation, the rapid termination of inhibitory response is critical so it does not interfere with the induction of LTP or LTD. Another experiment is the alternate control of projection neurons and interneurons in cortical areas, independent controls of neurons of direct and indirect pathways in the striatum to manipulate behavior.

(2) Simultaneous excitation and inhibition are not the same as non-excitation. The authors mentioned shunting briefly. Another possible issue is changes in osmotic balance. Activation of a Na+ channel and a Cl- channel will lead to net import of NaCl into the cell, possibly changing osmotic pressure. Please discuss.

We agree with the notion that osmotic, ionic and pH changes in small neuronal structure can be disruptive to the physiology and this is the reason we developed our approach where the fastest channelrhodopsins are used so we can minimize the channel opening time and the flux of ions through the channels when brief light illuminations are applied. Not only the flux of protons, sodium ions and calcium ions are minimized, the flux of chloride should be minimal as well (as the membrane potential should be close to the reversal potential of chloride reversal potential hence low ion flow). Hence our approach should be minimally disruptive compared to most other existing channelrhodopsin-based approaches when short or minimal light pulses were used in conjunction with our tools. This recommendation is included in the updated manuscript .

(3) The authors showed that in ZipT-IvfChr, orange light drives excitation and blue light does not. But what about simultaneous blue and orange light? Can the blue light overwhelm the effect of the orange light? Since the stated goal is to open the blue part of the spectrum for other applications, one is now worried about "negative" crosstalk. Please discuss and, ideally, characterize this phenomenon.

We now have performed this experiment. Simultaneous blue (470nm) and red light (635nm) stimulation does not produce AP (Fig .4 Suppl 1A). This suggests the inhibitory effect of ACR is more efficient than the excitatory effects of IvfChr due to their higher conductance, this re-emphasizes the rapid termination of the ACR effects is critical for minimal disruption of physiological effects in such pairing strategy.

(3.1) Does the use of the new tool require careful balancing of the expression levels of the ZipT and the IvfChr? Does it require careful balancing of blue and orange light intensities?

As with any optogenetic tool, the users should validate the efficacy of the tool in their own system. Our tool solely relies on the balanced expression of the 2A system, the efficiency of the two opsins and their degradation of the time-span of expression. These aspects of the tool would be better addressed in future versions of the tools or improvement of the BiPOLES-type of tandem expression in subsequent versions. From the instrumentation side, the light intensity and differential penetration depth requires careful consideration. However, this holds true in most optogenetic and fluorescence imaging-based approaches as well. In the current update of the manuscript, we have included further discussion on these aspects as well.

(3.2) Also, many opsins show complex and nonlinear responses to dual-wavelength illumination, so each component should be characterized individually under simultaneous blue + orange light.

We now have performed this experiment (please see our comment to point 3)

(3.3) I was expecting to see photocurrents at different holding potentials as a function of illumination wavelength for the coexpressed construct (i.e. to see at what wavelength it switches from being excitatory to inhibitory); and also to see I-V curves of the photocurrent at blue and orange wavelengths for the co-expressed constructs (i.e. to see the reversal potential under blue excitation). Overall, the patch clamp and spectroscopic characterization of the individual constructs was stronger than that of the combined constructs.

We have added the IV curves for the co-expressed construct at different holding potentials for 470nm and 635nm wavelengths. This shows reverse potential for the two wavelengths that are intended for in vitro and in vivo applications. Performing a similar experiment for a variety of wavelengths would not be as valuable, in part, due to the enormous amount of data generated. As we have shown in the study, the response of any channelrhodopsins vary with different light duration and light intensities in addition to the wavelengths and holding potentials. The results for each recorded cell could include stimulation by different wavelengths, stimulation by different illumination intensities, stimulation with different light duration in addition to different holding potentials. Not only would the results be highly variable from cell-to-cell, there will be potentially hundreds or thousands of combinations to be tested per cell (e.g., 5 light intensities @1, 2.5 , 5 , 10 and 20 mW/mm>sup>2, 8 different wavelengths @ 450nm, 475nm, 500nm, 525nm, 550nm, 575nm, 600nm and 625nm, 7 light durations @ 1ms, 5ms, 10ms, 50ms, 100ms, 500ms and 1s, and , and 6 holding potentials @ -80mV, -70mV, -60mV, -40mV, -20mV and 0mV would result in 1680 stimulation conditions per recorded cell).Technically, the significant lowering of membrane resistance when both IvfChr and ZipACR variants are activated simultaneously would compromise the quality of voltage-clamping even in HEK293 cells with series resistance compensation. We have yet to see any other studies that had included such ambitious electrophysiology experiment for the channelrhodopsin characterization, likely due to the feasibility of such experiment.

**Reviewer #3 (Public Review):**
(1) The enhanced vf-Chrimson could potentially be a highlight of the manuscript, serving broader applications. Yet, gauging the overall improvements of ivf-Chrimson in comparison to other Chrimson variants remains intricate due to several reasons. First, photocurrents from ivf-Chrimson seem smaller than those from C-Chrimson (Supplemental Figure 3), and a direct comparison with standard vf-Chrimson is absent.

We appreciate the reviewer’s positive view of our modified variant. We did not emphasize this particular modification as it was identical to our previous published modification and similar to that previously published by others (CsChrimson and C1Chrimson). In all these cases, improved membrane expression was consistently detected. We believe that expression data and our comparison of C-Chrimson and IvfChr is sufficient to justify the improved membrane expression and function.

Second, while membrane expression of ivf-Chrimson appears enhanced in provided brightfield recordings, the quantitative analysis would necessitate confocal microscopy and a membrane marker (Supplemental Figure)

We have now quantified the results with a membrane palmitoylated mCherry using confocal microscopy shown in Fig 2 Suppl1 A. We measured the Pearson Correlation Coefficient of the mCherry with EGFP or Citrine signal for the 6 constructs (vfChrimson, vfChrimson with trafficking sequence, vfChrimson with N-terminal signaling peptide from oChIEF (C-vfChrimson), vfChrimson with trafficking sequence and N-terminal signaling peptide from oChIEF (IvfChr), BiPOLES with EGFP or citrine and vfChrimson) and the results were identical and consistent with the prior results using epifluorescence microscopy.

(2) Finally, other N-terminal modified Chrimson variants, like CsChrimson by Klapoetke et al. in 2014 and C1Chrimson by Oda et al. in 2018, have been generated. Comparing ivf-Chrimson to vf-CsChrimson or vf-C1Chrimson would be important to evaluate the benefits of the applied N-terminal modification.

Our development of IvfChrimson is similar to the approach of vf-CsChrimson and identical to that of vf-C1Chrimson and we do not claim these modifications to be unique or superior. However, we have developed our design independently of these other studies and we have more extensive functional comparison and characterization data of our IvfChrimson variant than the other studies.

(2.1) The action spectra of ZipACR suggest peak absorption of ZipACR WT and its mutant at 525 - 550 nm (Fig. 3). This is even further red-shifted than previously reported by Govorunova et al. Further action spectra recordings differ for all constructs between recordings initiated with blue or red light (Supplementary Fig. 5). This discrepancy is unexpected and should be discussed.

We thank the reviewer for the comment, this was a mistake in the traces used for the figure. The example traces were the spectral response measured from the 400 nm to 650 nm instead of the 650 nm to 400 nm order shown in the spectral data. This has now been corrected.

Additionally, the representative photocurrents of Zip(151V) in Fig. 3D1 do not align with the corresponding action spectrum in Fig. 3D2 as they show maximal photocurrents for 400 nm excitation.

Please, see point above.

(3) The authors introduce two different bicistronic expression cassettes-ZipT-IvfChR and ZipV-IvfChR-without providing clear guidelines on their conditions of use. Although the authors assert that ZipT is slower and further red-shifted than ZipV, the differences in the data for both ACR mutants are small and the benefits of the different final constructs should be explained.

In our testing in neurons, ZipT has less ‘escaped’ spikes after the termination of the light pulses in the cells we have tested. However, this is dependent on the membrane properties such as capacitance and resistance of the cells. ZipV has a faster termination time and in some situations may be necessary due to its faster termination time and reduced disruption of physiological processes.

We have now included this discussion in our updated manuscript.

(4) The ZipT/V-IvfChRs are designed as bicistronic constructs; yet, disparities in membrane trafficking and protein degradation between the two channels could lead to divergences in blue and red light photoresponses. For future applicants, understanding the extent of expression ratio variations across cells using the presented expression cassettes could be of significance and should be discussed.

We now have included this discussion in our responses above.

**Reviewer #1 (Recommendations For The Authors):**
(1) The Figure 1a mV cartoon traces for chloride are confusing. The chloride currents are depolarizing, not hyperpolarizing. As noted by the authors, these channels largely generate AP blockade through shunting inhibition (division), not hyperpolarization (subtraction).

The figure has been corrected.

(2) Figure 2A does not show where the light is applied. Why are some of the bars blue and some of them not filled?

This has been corrected

(3) Figure 2C1 does not show where the light is applied. There should be an inset to detail the blue-light-cessation-evoked AP. Also doesn't give the holding potential.

The requested details are added.

(4) Figure 2C2 inset is described as showing that "Light-induced currents with 470 nm illumination were initially outward but turned inward immediately following light offset." Is that correct? It looks to me like the current turns inward about half-way through the light pulse and then becomes even stronger after the light turns off. That is also consistent with the CC traces, which appear to show a transition toward depolarization during the light pulse before the AP initiation at light offset.

Yes, the reviewer's observation is correct. There are blue light-induced outward and inward current peaks at the onset and offset of the light. Accordingly, we have modified the phrasing for Fig. 2C2.

(5) Figure 3D1 shows that Zip(151V) has a peak current at 400nm, with a steady increase in current from red to blue, however, this is not the case in the summary data in 3D2. It's also not shown in Supplementary Figure 5B. What's going on?

We apologize for the prior version of the figure associated with the first submission. The example traces from 400nm -> 650 nm were incorrectly included in the figure whereas the 650nm -> 400 nm example traces should be included. This has been corrected.

(6) Figure 3D1 has no time scale.

It is now been included

(7) Figure 3E1 should read "Transduced" and not "Transfected"

This has been corrected.

(8) IvfChr fidelity drops off dramatically at 20hz...down to 50% efficiency of generating APs. This is described in the legend as "high frequency". Maybe the cart came before the horse in this figure...as it looks like in panel C that using less light power density improves fidelity in the dual opsin configuration with red light stimulation...why not use that power for the characterization? Did you try any higher frequencies? Or longer pulse widths? This is an important characterization to inform further use of the tool. This shortcoming isn't a cell-intrinsic limitation, as the 470nm stim with IVfChr was 100% successful at both 10hz and 20hz.

It is known that red but not blue light pulses induce desensitization (optical fatigue) in red-shifted ChR variants. Indeed, one can reinstate the response to red light, by giving violet-blue light pulses (Fig 4. Suppl 2). We think this is the reason that the 470nm stimulation was more effective in inducing AP in cells expressing IvfChR. Higher light intensities induce greater desensitization, but are preferred for faster opening of channels and depolarization of neurons. This can explain why, in some situations, lower light intensities were more effective in producing APs when pulse trains were used. We have recordings from cells firing APs at 40Hz (not included). All these cells had high expression levels of the opsin.

(9) Figure 4D: why use 100ms pulse width? How do you know that this isn't causing depol block? Or some of the nefarious concerns that are raised in the discussion, such as "...disrupt[ion of] normal neuronal physiology and signal processing that occurs in millisecond time scale"?

We used 100ms pulse duration to follow the published protocol that this experiment is based on (Lin et al., 2013, Nature Neuroscience).

(10) Figure 4E-bottom: What is the blue peak at light onset? Is the tool driving early activation before silencing?

There seems to be an early, sharp and brief activation by blue light. We don’t know the definite cause of this, but we speculate this is driven by blue-light activation of ZipACR and not the IvfChr portion of the construct. The reason is that such a sharp rise is absent when only IvfChr is expressed (Fig. 4E, upper panel). Soma-targeted motif tethered to channelrhodopsins is known to result in preferential expression of channels close to soma but does not exclude the expression of channelrhodopsin in axonal and dendritic compartments, especially when animals are allow to recover for long period of time after viral injection. We believe that ZipACR at axonal terminals where the chloride concentration is high can still cause blue-light evoked depolarization and transmitter release. We observed this phenomenon in two mice in their first trial. The data for individual trials for each mouse are included in a supplementary table.

(11) Figure 4G: Earlier in this same figure (B2, C), 470nm light was more effective at stimulating IvfChr than 635nm light. Is it unexpected that 638nm light would in this in vivo context be more effective at driving IvfChr responses than 450 nm light (at least as reflected by the AUC measurements)? Does this reflect fiber placement and light penetration/scattering?

The spectral peaks of Chrimson-based variants including vfChrimson are all centered around 600 nm, and at 635 / 638 nm light, the amplitudes of photo-response decline, the channel onset slows, and the channels suffer greater desensitization. In isolated preparations where the light penetration is similar between 635 / 638 nm and 470 nm, 470 nm responses can outperform 635 / 638 nm responses due to its lack of desensitization and higher consistency in its response. This is also a strong reason that we have developed our current approach. In *in vivo* preparation shown in Fig. 4D-G, the much higher tissue penetration of 638nm light due to reduced absorption and reduced scattering can offset the performance of IvfChr to 450 nm light.

(12) In the methods, it is noted that different viral batches appear to generate different levels of neuronal toxicity. If that is the case, how did you differentiate between true differences between constructs vs. differential cell health effects?

For figure 4D-F (whisker movement), we determined virus toxicity using NeuN staining. In slice recordings, we used the electrophysiological property of the neurons to assess their health. For this manuscript, we had one batch of virus that produced toxicity. We did not include any data from this batch.

**Reviewer #2 (Recommendations For The Authors):**
● Define AUC on first use.

It is now defined.

● Figure 3C2: Please explain how the photocurrents were normalized. As presented, it looks like under strong orange light, the ZipACR has higher photocurrent than the ivfChr.

This is due to the fact vfChrimson and other Chrimson-based variants do not fully recover in the dark after 590 nm stimulation. We tested IvfChrimson with both reconditioning light pulse of 405 nm and without 405 nm and we can consistently reach a greater ‘maximal’ response from the same cell after 405 nm reconditioning (see Fig. 4 Suppl 2). We therefore normalize the response to the maximal recorded response of the cell often achieved with 10 or 20 mW/mm^2^ 590 nm stimulation after 405 nm reconditioning. We understand this can be confusing and have now replaced the light-intensity response in Fig. 3C2 with the one with 405 nm reconditioning which is easier to interpret for the readers.

● P. 3: "As expected, blue light pulses induce transient membrane suppression..." Unclear what "suppression" means. Shunting? Hyperpolarization?

We rephrased this to “As expected, blue light pulses transiently suppress APs…”

● P. 3: "illumination at 470 nm and 590 nm wavelengths led to similar amounts of courtship song (110.1 {plus minus} 12.8 and 78.5 {plus minus} 11.6,n = 16-17, respectively)". What are the units of "courtship song"?

The unit for courtship song is the number of pulses per 10 seconds. This has been clarified in the figure.

● P. 5: The quantification of photocurrent in terms of pA/pF/A.U. is non-standard. I understand the impetus to normalize by expression to give something proportional to per-molecule conductance, but a user cares about overall photocurrent. Please also give the real photocurrents, either pA or pA/pF.

We have provided the real photocurrent in pA or pA/pF where scientifically appropriate. To avoid selection and experimenter’s bias in our data, we did not set criteria for data elimination for cells with specific fluorescence intensity or photocurrent amplitude. Some resulting response can range from vary up to 20 folds from the same construct in many experiments. We do not believe that averaging absolute photocurrent amplitude would be justified due to the imbalance of weighing in the results. We do acknowledge that not selecting or eliminating data points would introduce higher noise in recordings with smaller responses but this is preferable over the selection or experimenter bias that is likely to be introduced otherwise.

● Please quote illumination intensities wherever possible.● P. 7: why was the red light crosstalk into Zip(151T) tested at 635 nm instead of 590 nm? Isn't the relevant parameter 590 nm, since that will be used for the excitatory opsin?

In all our characterizations of the constructs using slice electrophysiology recordings, we used 635nm instead of 590nm. The reason is that compared to 590nm wavelength, at 635nm the photocurrent for Zip(151T) and Zip(151V) is significantly reduced (Fig. 3D1,D2).

● P. 10: "we examined the power at which responses to 470 nm and 635 nm lights induce APs in neurons expressing ZipT-IvfChr, ZipV-IvfChr, or IvfChr", but the preceding sentence says you didn't test the ZipT-IvfChr. This is confusing, please clarify.

The previous paragraph refers to the photocurrent recordings in HEK293 cells where our fast LED based illumination system is limited to 590 nm light, whereas the subsequent paragraph refers to the brain slice neuronal recordings. We have now emphasized the difference of the experiments in the rewrite.

● Fig. 4B1, top: Why don't the blue traces return to the same baseline after the stimulus epochs?

We observed this shift in baseline (~4mV more depolarized) in cells expressing IvfChR (or vfChR) only with blue light stimulation. This was observed in the neurons recorded in the CA1 as well (data not shown). There was no such a change following red light stimulation (Fig. 4B1). Therefore, this should not affect the applicability of our construct. The original paper introducing vfChR did not test the responses of their constructs to blue light. There could be another photocycle state that is activated stronger by 470nm than 590nm and it has a slow off-rate, but this is only a speculation from our side. It must be noted we did not observe such a phenomenon in cells expressing ChrimsonR (Fig. 1 Suppl 1C).

● Fig. S3B, right: The two colors are barely distinguishable on the graph. Consider more distinct colors and/or different symbols.

It has been changed accordingly.

● P. 15: "However, we do not recommend the use of orange light pulses, as we observed a significant photocurrent in this wavelength." Not clear what this is referring to. Which construct? Under which circumstances shouldn't one use orange light pulses? Where's the data showing this?

This is referring to Fig. 3D1,D2 and Figure 4 suppl Fig. 2 which show a normalized ~40-50% photocurrent at 590nm. Now in the text, the reference figures for the data are added.